# Metabolic reprogramming by Zika virus provokes inflammation in human placenta

Qian Chen [1], Jordi Gouilly [1], Yann J. Ferrat[2], Ana Espino[1], Quentin Glaziou[1], Géraldine Cartron[3], Hicham El Costa [1], Reem Al-Daccak[4] & Nabila Jabrane-Ferrat [1]✉

The recent outbreak of Zika virus (ZIKV) was associated with birth defects and pregnancy loss when maternal infection occurs in early pregnancy, but specific mechanisms driving placental insufficiency and subsequent ZIKV-mediated pathogenesis remain unclear. Here we show, using large scale metabolomics, that ZIKV infection reprograms placental lipidome by impairing the lipogenesis pathways. ZIKV-induced metabolic alterations provide building blocks for lipid droplet biogenesis and intracellular membrane rearrangements to support viral replication. Furthermore, lipidome reprogramming by ZIKV is paralleled by the mitochondrial dysfunction and inflammatory immune imbalance, which contribute to placental damage. In addition, we demonstrate the efficacy of a commercially available inhibitor in limiting ZIKV infection, provides a proof-of-concept for blocking congenital infection by targeting metabolic pathways. Collectively, our study provides mechanistic insights on how ZIKV targets essential hubs of the lipid metabolism that may lead to placental dysfunction and loss of barrier function.

[1] Center for Pathophysiology of Toulouse Purpan, CNRS - Inserm - University of Toulouse, 31024 Toulouse, France. [2] CERAG Laboratory, University of Grenoble Alpes, 38040 Grenoble, France. [3] Department of Obstetrics and Gynecology, Paule de Viguier Hospital, 31059 Toulouse, France. [4] Saint-Louis Hospital, Inserm UMRS976, University of Paris Diderot, 75010 Paris, France. ✉email: nabila.jabrane-ferrat@inserm.fr

The mosquito-borne Zika virus (ZIKV) is a single strand positive-sense RNA virus that belongs to the *Flaviviridae* family. ZIKV infection is mostly asymptomatic but in early pregnancy it has been linked to pregnancy loss and devastating birth defects including the life-threatening fetal brain abnormalities referred to as congenital ZIKV syndrome[1,2]. The replication of ZIKV in a wide range of fetal and maternal cells prompted the idea that maternal–fetal interface can serve as a replication platform enabling viral amplification before dissemination to the fetus[3,4]. However, despite intense investigation, mechanisms driving placental dysfunction, and subsequent ZIKV-mediated fetal pathogenesis are not fully understood.

Lipids are highly diverse cell components that play a central role in maintaining appropriate cellular functions, including membrane structure, energy sources, and signal transduction. Alteration in lipid metabolic pathways is a leading cause of many human diseases[5,6]. The fetal placenta is an autonomous organ endowed with an extraordinary high lipid content and metabolic rate to support fetal development. Mounting evidence links alteration of the placental lipid metabolism to the etiology of many "great obstetrical syndromes" including gestational diabetes mellitus (GDM), miscarriage, congenital disorders, fetal growth restriction (FGR), and pre-eclampsia[7–9].

Lipid droplets (LDs) are fat storage organelles derived from the endoplasmic reticulum (ER) membrane under conditions of fatty acids excess. In contrast to other cellular organelles, LDs are composed of a neutral lipid core surrounded by a monolayer of phospholipids (PLs) harboring coat proteins and lipid metabolism enzymes[10]. The ER-resident diacylglycerol acyltransferase 1 (DGAT1) is central for LD biogenesis[11,12]. LDs make contact with many organelles to supply necessary lipids for energy production, membrane biogenesis, and intracellular vesicle trafficking. LDs also act as regulatory hubs to prevent lipotoxicity and maintain lipid homeostasis. The impairment of their protective cellular response has been associated with metabolic disorders[13]. Despite differences in their transmission mode, single-stranded positive RNA viruses hijack the ER membrane network and subvert lipid homeostatic pathways to build specific endomembrane organelles for viral replication (ROs). Both viral and host factors are supposed to be concentrated in ROs to facilitate assembly and shield nascent virions from immune assaults[14,15]. Increased knowledge about virus–host interactions and the role of host lipid metabolism prompted the development of therapeutic strategies that have been proven effective alternatives in controlling viral pathogenesis in many model systems[16,17].

Lipids are also a repository of potent bioactive mediators, such as eicosanoids. Eicosanoids are derived from long-chain polyunsaturated fatty acids (PUFAs) through a complex pathway[18]. Similar to cytokines, bioactive lipid mediators (LMs) constitute a finely tuned and complex lipid signaling network that regulates homeostatic and inflammatory processes. Whilst some LMs have been implicated in the control and clearance of viral pathogens[19,20], it remains unclear how ZIKV infection would affect the biosynthesis of placental lipid metabolites and perturb the homeostatic equilibrium of the placental barrier. Given the central role of lipids in fetal and placental development, dysregulation of this signaling network is very likely to contribute to placental inflammation and adverse pregnancy outcomes[21,22]. Unraveling such a mechanism would open new avenues for therapeutic strategies to prevent congenital ZIKV syndrome.

In this study, we used large-scale quantitative metabolomics to investigate the impact of ZIKV on human placenta during early pregnancy. We demonstrate that ZIKV reprograms the placenta lipidome to accommodate viral life cycle. We also provide evidence that loss of metabolic homeostasis is associated with mitochondrial dysfunction and imbalance in the pro-/anti-inflammatory equilibrium that characterize severe pregnancy outcomes. Our findings uncover a potential mechanism by which ZIKV overcomes the barrier function of the fetal placenta and may have important implications for the development of therapies for a wide range of placental diseases.

## Results

**ZIKV infection adjusts placental neutral lipids**. Metabolic reprogramming is a well-recognized hallmark of human disease including the great obstetrical syndromes linked to placental dysfunction[9,23]. Although congenital ZIKV infection can presumably occur at different gestational ages, severe sequelae have been linked to infection during early pregnancy[1,24,25]. To determine whether ZIKV perturbs the metabolic status of the fetal placenta, we therefore used first trimester pregnancy samples. Placentas were challenged with the Asian strain (KU886298) of ZIKV at $6 \times 10^{10}$ RNA copies/mL (equivalent to a MOI of 1) obtained from the second passage in Vero cells[3,26–28]. The kinetic of infection was followed by quantification of viral RNA in culture supernatants. The maximum increase of ZIKV RNA was reached within 5 days of infection (Fig. 1a). We next subjected matched mock- and 5 day infected whole tissue samples to large-scale quantitative shotgun lipidomic coupled to mass spectrometry analyses. We analyzed four major groups of neutral lipids: cholesterol, diacylglycerols (DG), cholesterol ester (CE) and triacylglycerols (TG). Cholesterol occupies more than 65% of total placenta neutral lipids whereas aggregated DG and TG account only for ~30% (Fig. 1b). ZIKV-infected placentas displayed significantly higher amounts of total neutral lipids without changing their global distribution within different groups. Further analysis of independent groups revealed that the infection significantly increases the amount of DG and induces a tendency of higher TG and cholesterol contents without affecting CE (Fig. 1c–f). Given the importance of DG and TG in lipid homeostasis, we next examined the impact of viral infection on their related subspecies. Despite variability between donors, ZIKV infection significantly enhanced the relative abundance of subspecies with long acyl-chains containing 18 or more carbons in their tails, namely DG (18_18), DG (18_20), TG (18_18_18) and TG (18_18_20) (Fig. 1g, h). Hierarchical clustering using an algorithm that allows grouping of similar samples based on the pattern of relative abundance, heatmap, indicated that ZIKV-infected placentas nurture significant enhancement of differential neutral lipid species (Fig. 1i).

**ZIKV infection alters the biogenesis of PLs**. PLs are central for cellular membranes functionality since they provide adequate fluidity, vesicular traffic, molecular transport, and biosynthesis[29,30]. Phosphatidic acid (PA) and its product, DG, are the main precursors for PL synthesis through the Kennedy or CDP-DAG pathways (Fig. 2a, schematic summary). Therefore, we investigated the impact of ZIKV infection on placental PL metabolism. Matched mock- and ZIKV-infected Placentas were subjected to Bligh and Dyer's extraction followed by LC-MS/MS quantification. In human placenta, phosphatidylcholine (PC) constitutes ~63% of total placental PLs, whereas sphingomyelin (SM) represents only ~23% and other PLs species <10% (Supplementary Fig. 1). Compatible analysis revealed significant upregulation of the relative abundance of PC, phosphatidylethanolamine (PE) and phosphatidylinositol (PI) species in ZIKV-infected samples (Fig. 2b–d). Although it did not reach significance, the phosphatidylserine (PS) group was also increased by the infection (Fig. 2e). No effect was observed for ceramide (Cer) or SM subspecies (Supplementary Fig. 2).

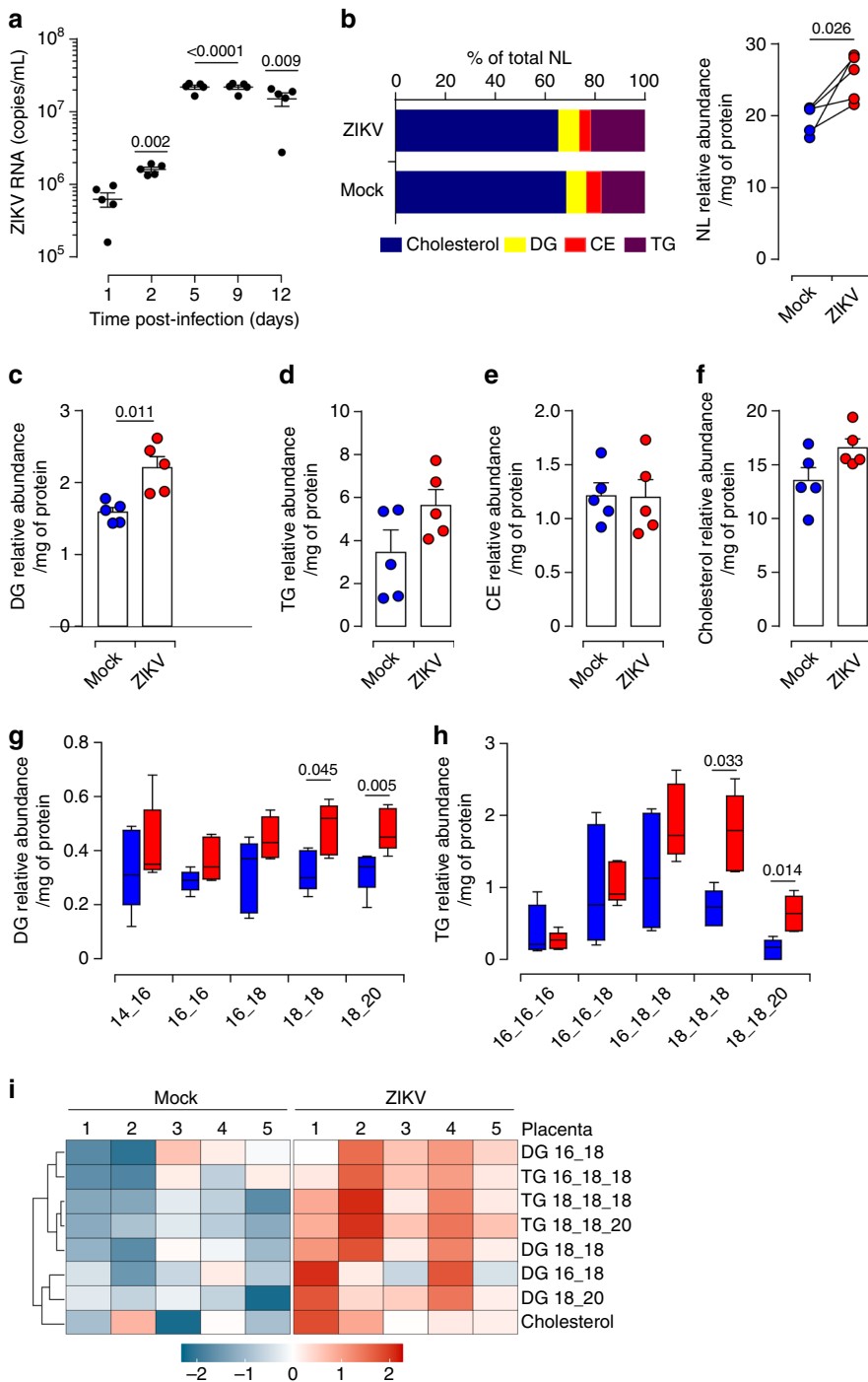

**Fig. 1 ZIKV infection impairs placental neutral lipid metabolism. a** Culture supernatants are collected from five independent ZIKV-infected first trimester placentas following the indicated time. Viral load is determined by qRT-PCR. **b** Neutral lipids (NL) are extracted from matched mock- and ZIKV-infected first trimester placentas 5 days post-infection (dpi). Four major neutral lipids, cholesterol, diacylglycerol (DG), cholesteryl ester (CE) and triglyceride (TG), were determined in human placenta. Quantification of neutral lipid species was achieved by comparison of the peak area of individual species relative to their class-specific internal standards. Values are normalized based on the protein level for each sample. Relative abundance of NL is calculated as the sum of DG, TG, CE, and Cholesterol amounts. **c–f** Quantification of DG, TG, CE, and cholesterol. Relative abundance of DG, TG, and CE represents the total abundance of the different subspecies in each group. **g–h** Quantification of DG and TG subspecies characterized by the number of carbon atoms in each acyl chain. Mock samples are depicted as blue bars and ZIKV-infected samples as red bars in the graph. Boxplot: boxplot medians (center lines), interquartile ranges (box ranges), whisker ranges. **i** Heatmap representing relative abundance profile of NL subspecies. Each column represents a single donor and each row corresponds to a single sub-specie of neutral lipids. All the data sets are presented as mean values ± SEM of first trimester placentas from five independent donors. *P* values are computed using paired two-tailed Student's *t* test. Source data are provided as a Source Data file.

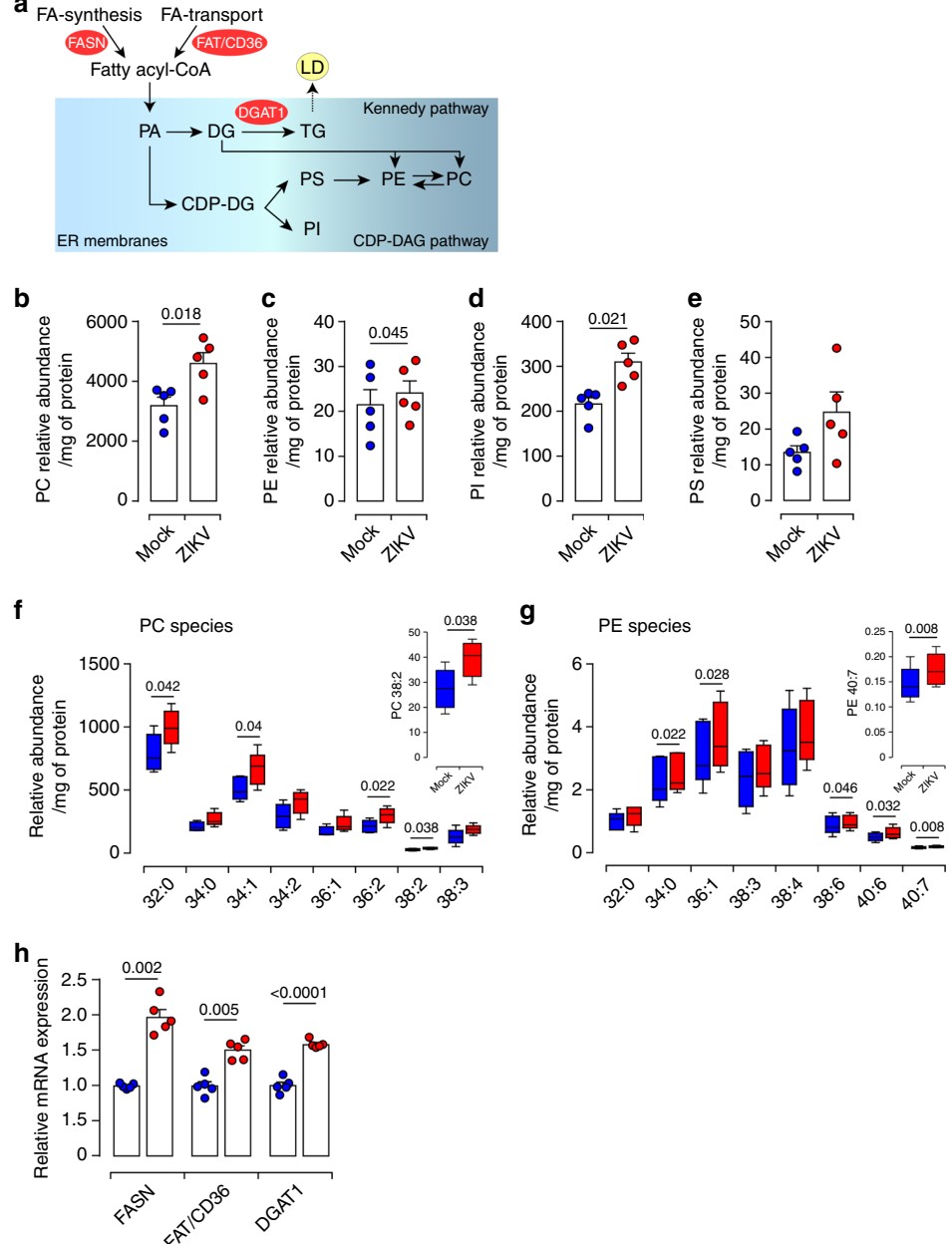

**Fig. 2 ZIKV infection perturbs the phospholipid placental profile. a** Simplified schematic overview of the phospholipid (PL) metabolism pathways. Fatty acid (FA), phosphatidic acid (PA), cytidine diphosphate diacylglycerol (CDP-DAG). Lipid droplet (LD). Fatty acid synthase (FASN) catalyzes FA de novo lipid synthesis. The transporter FAT/CD36 assists in FA uptake from the exogenous environment. The ER-resident diacylglycerol acyltransferase 1 (DGAT1), an essential enzyme for LD biogenesis, catalyzes the final step in TG biosynthesis. **b–g** PLs were extracted from mock or ZIKV-infected placenta explants 5 dpi and subjected to LC-MS/MS analysis. Quantification of PL species was achieved by comparison of the peak area of individual species relative to their class-specific internal standards. Values are normalized based on the protein level for each sample. The relative abundance of PL species is calculated as the sum of subspecies within different subgroups. **b** Phosphatidylcholine (PC), (**c**) phosphatidylethanolamine (PE), (**d**) phosphatidylinositol (PI), (**e**) phosphatidylserine (PS). **f–g** Relative abundance of differentially altered phospholipid subspecies from PC (**f**) and PE (**g**). Boxplot: boxplot medians (center lines), interquartile ranges (box ranges), whisker ranges. **h** Total RNA extracted from matched mock- and ZIKV-infected first trimester placentas at 5 dpi. The mRNA expression levels were determined by qRT-PCR. Mock-infected placenta (blue) and ZIKV-infected (red). Results are presented as mean values ± SEM of first trimester placentas from five independent donors. *P* values are computed using paired two-tailed Student's *t* test. Source data are provided as a Source Data file.

Further profiling of individual subspecies revealed PC32:0 as the most abundant among 19 molecular subspecies detected in the PC group and identified a total of 16 molecular subspecies within the PE group with much lower abundance compared to the PC group (Supplementary Fig. 3). Most of ZIKV-increased PC and PE subspecies displayed long-chain fatty acids with unsaturated structure (C32–C40) (Fig. 2f, g).

In a given membrane, even slight alterations to PL content can significantly change physical properties including spontaneous curvature, fluidity and permeability[31,32]. In this context, the

observed alterations in the biogenesis of individual PL subspecies suggest that ZIKV, through reprogramming of the lipid metabolome, may crucially impact the architecture of placental membranes increasing fluidity and permeability.

**ZIKV impairs placental lipid homeostasis**. To determined which pathways are responsible for ZIKV-induced placental metabolic alterations, we investigated expression levels of three essential enzymes: FASN (fatty acid synthase) which is central for de novo synthesis of fatty acid (FA), FAT/CD36 (fatty acid translocase), which facilitates FA transport and DGAT1, which is involved in LD biogenesis (Fig. 2a). Our analysis revealed that *FASN* and *FAT/CD36* mRNA levels are significantly increased in ZIKV-infected placentas compared to mock-infected samples (Fig. 2h).

The increase in *FASN* and *FAT/CD36* expression will enhance de novo synthesis of FAs and facilitate the transport of long-chain FAs from extracellular media for placental lipogenesis. Similar to *FASN* and *FAT/CD36*, we also detected significant increases of *DGAT1* mRNA levels in ZIKV-infected placentas which can promote TG synthesis. Our observations illuminate diverse pathways that ZIKV exploit to satisfy lipid needs namely FA synthesis and transport, and storage in LDs.

**Placental LDs are tangled during ZIKV Infection**. To provide insights into a possible role of LDs during ZIKV infection, we examined the distribution and morphology of these organelles in ZIKV-infected placenta. LDs were visualized in placental thin sections with the BODIPY 493/503 dye which stains the LD core. Trophoblast cells were visualized by immunostaining of cytokeratin 7 (CK7). ZIKV infection was detected with antibody against the NS3 viral protein. We found that small size LDs were distributed throughout mock-infected tissue sections, both in CK7 positive and negative cells. In contrast, the majority of LDs accumulates around infected focal sites within ZIKV-infected placental villi (Fig. 3a). In addition, large amounts of LDs were found in uninfected neighboring cells.

To further characterize the effect of ZIKV on placental LDs, we performed immunohistochemistry (IHC) on frozen thin sections from mock- or ZIKV-infected placentas. LDs were visualized with Oil Red O (ORO) and nuclei with hematoxylin staining (Fig. 3b). Our analysis showed that ZIKV infection induces large amounts of LDs with a nearly 1.7-fold increase in global distribution (Fig. 3c). The increase in LD distribution was also associated with major changes of the placental tissue architecture illustrated by morphometric alterations to the stroma and syncytiotrophoblast (STB) outer layer of the villi (Fig. 3b). Taken together, these findings demonstrate that ZIKV induces upregulation of the placental LD biosynthesis and further provide meaningful insights into the enhancement of DG and TG subspecies during the infection.

**ZIKV induces LD accumulation and morphological changes**. To gain further insights into the mechanisms that underlie changes in LD morphology during ZIKV infection, we analyzed quantities and size distribution of LDs at the cellular level. Viral infection was visualized with anti-Flavivirus envelope group antigen antibody (anti-Env) or anti-NS3 antibody staining (Fig. 4 and Supplementary Fig. 4). LDs were labeled with the ORO or BODIPY 493/503 dye. ZIKV-infected cells accumulated large numbers of LDs with a significantly larger morphometric area (Fig. 4a and Supplementary Fig. 4a). Both the number and average volume of LDs were significantly increased by the infection (Fig. 4b and Supplementary Fig. 4b). Further stratification of LDs according to their size distribution and evaluation of the percentage in each diameter range revealed an average size

of approximately 1 μm, although other sizes were also observed. Compared to mock-infected cells, ZIKV infection significantly decreased the percentage of small size LDs (<1 μm) and increased the percentage of large LDs (>1.5 μm) (Supplementary Fig. 4c). The shift in size with the accumulation of larger size organelles was even greater 5 day post-infection (dpi) (Fig. 4b, right panel). These findings indicate that ZIKV infection changes LD dynamics with respect to their size and number. This phenomenon is triggered either by fusion of several LDs or by organelle growth through integration of excess lipogenesis. We also observed an increase of the LD amount and size in uninfected neighboring cells suggesting a bystander effect.

In addition to trophoblast, the unique cell type of the placenta, the villous core contains mesenchymal/fibroblastic-like stem cells that contribute to the global architecture[33]. To determine whether a specific cell type support modulation of LD morphogenesis, we performed co-immunofluorescence staining of viral proteins, CK7 and LD. Morphological analysis indicated that ZIKV infection induces the accumulation of large size LDs in both trophoblast and stromal cells (Fig. 4c, d and Supplementary Fig. 4d). Three-D reconstitution model revealed that most of the large size LDs aggregate close to ZIKV viral particles, as evidenced by anti-Env staining (Fig. 4c, d, lower panels). Consistent with immunofluorescence, ultrastructural analysis using transmission electron microscopy (TEM) depicted abundant accumulation of LDs near the ER sheets in ZIKV-infected cells relative to mock-infected cells (Fig. 4e). Within the ER lumen, we detected LD surrounded by a large number of viral particles and connected with mitochondria (Fig. 4e, right panel). The juxtaposition of LDs with viral particles further highlights their involvement in the virus life cycle. Furthermore, the close connection between LD and mitochondria is likely to facilitate material exchange between these key organelles. Thus, ZIKV infection occurs in trophoblast and stromal cells and likely impairs LD morphology to support viral genome replication and/or assembly of progeny virions.

**ZIKV modulates LDs in both infected and bystander cells**. Since bystander effects during pathogen infections underlies progression to advanced diseases, we extended our investigations to uninfected neighboring cells. We, therefore, examined the spatial distribution of LDs in uninfected neighboring placental cells at 3 and 5 dpi. LD content was determined by immunofluorescence in Env+ infected cells and Env− cells. Neighboring cells harbored a considerable number of LDs which is even higher than that of infected cells (Fig. 5a and Supplementary Fig. 5a). Further quantification analysis showed that these bystander cells did not only contain higher amounts but also larger size LDs than infected cells expressing the viral envelope (Fig. 5b). Increased LD morphometric parameters in bystander cells were also observed at early infection time points (3 dpi) (Supplementary Fig. 5b).

Bystander effects of a variety of viruses implicate the establishment of intercellular channels within the gap junction, which play a central role in coordinating metabolic changes of neighboring cells, but also can implicate soluble mediators secreted by infected cells[34–36]. We thus tested this latter possibility by culturing freshly isolated placental cells with UV-irradiated conditioned medium harvested from ZIKV-infected placental cells (UV-ZIKV). Conditioned media from mock-infected cells were UV-irradiated and used as control (UV-Mock). Microscopic and quantitative analysis indicated remarkable LD accumulations in cells treated with UV-ZIKV compared to those treated with UV-Mock. Both the amount and the size of LDs were significantly increased in the presence of UV-ZIKV treated samples (Fig. 5c, d).

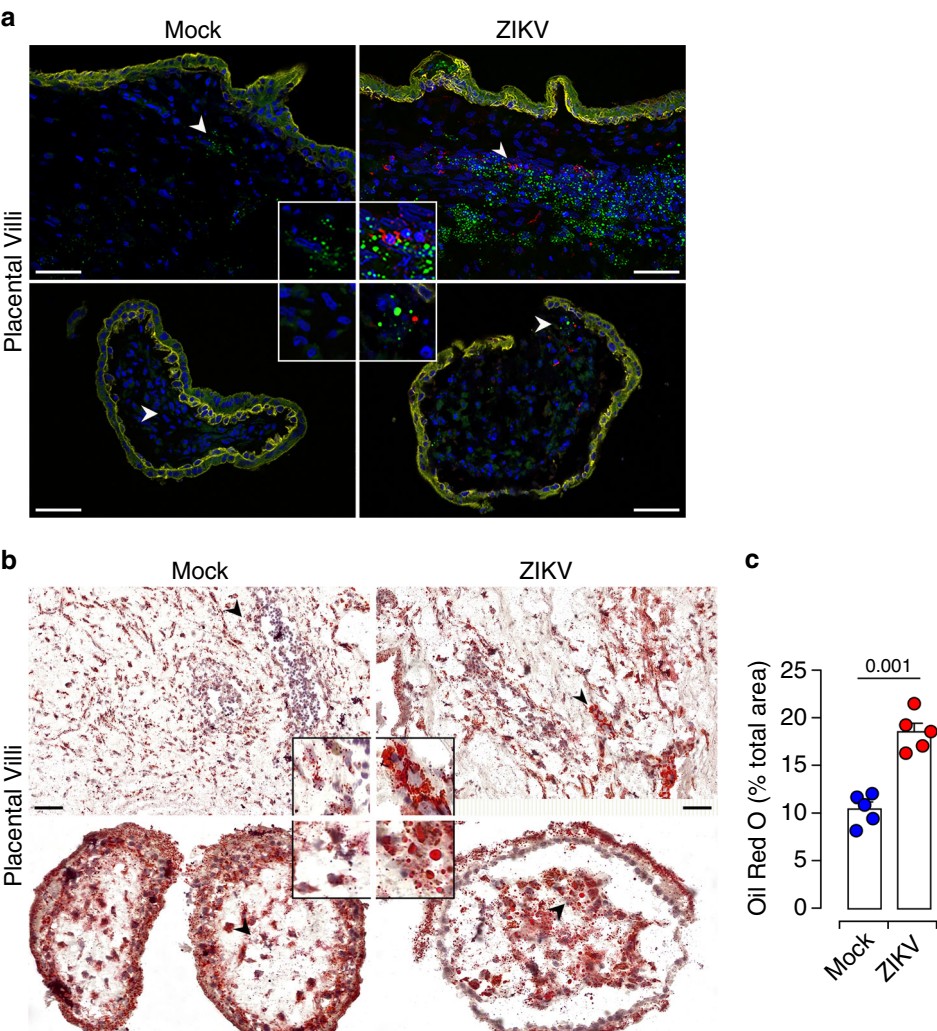

**Fig. 3 Accumulation of LDs in ZIKV-infected placenta. a** Representative thin sections of matched mock and ZIKV-infected placentas 5 dpi. Immunostaining is performed with anti-NS3 (red) and anti-CK7 (yellow) antibodies. LDs are stained with the BODIPY 493/503 dye (green) and nuclei with DAPI (blue). High magnification of the indicated area (arrowhead) is shown in insets. Scale bar, 50 μm. Data are representative of at least three independent experiments. **b** Oil Red O staining of neutral lipid in tissue sections from matched mock- and ZIKV-infected placenta (red). Nuclei are counterstained with hematoxylin. High-magnification images of the indicated area (arrowhead) are shown in insets. Scale bar, 50 μm. **c** Quantification of Oil Red O staining by Image J showing increased staining area in the infected placental sections. All data are presented as mean ± SEM of first trimester placentas from five independent donors. *P* values are computed using paired two-tailed Student's *t* test. Source data are provided as a Source Data file.

Thus, ZIKV infection does not only directly regulate the lipid metabolism to support the virus life cycle but also induces bystander effect through gap junction exchange of cellular components or paracrine pathways, which may further emphasize the critical role of LD alterations in viral pathogenesis.

**ZIKV exploits LDs for genome replication.** The ER-resident DGAT1 enzyme, which is essential to LD biogenesis, plays a central role in the formation of Hepatitis C (HCV) particles[37,38]. We, therefore, hypothesize that ZIKV-induced morphometric changes of LDs are necessary for the virus life cycle. To test the involvement of these lipid regulatory hubs in viral replication, we used the pharmacological inhibitor A922500 (DGAT1i) to block the enzymatic activity of DGAT1 and inhibit the conversion of FAs to TGs (Supplementary Fig. 6). Freshly isolated placental cells were treated with different concentrations of DGAT1i for 5 days. As expected, immunofluorescence analysis using the BODIPY 493/503 dye demonstrated that compared to DMSO

control, DGAT1i treatment of placental cells substantially reduced the LD amounts without impairing the cell viability even in the presence of the highest concentration (5 μM) (Supplementary Fig. 6b, c). Next, we examined the impact of DGAT1i during ZIKV infection. To this end, placental cells were challenged with ZIKV in the presence of different concentrations of the inhibitor. The spatiotemporal distribution of LD was analyzed 5 dpi by immunofluorescence (Fig. 6a, b). Compared to DMSO control, cells treated with DGAT1i harbor only scarce LDs. Further morphometric quantification revealed that both the number and average volume of LDs are significantly reduced in the presence of DGAT1i. The impairment of LD dynamics was associated with a lower infection rate.

To further ascertain the role of LDs in ZIKV life cycle, we investigated whether inhibition of LD formation impacts infectious particle production. We determined the production of infectious particles using plaque assay (Fig. 6c, d). Consistent with our immunofluorescence analysis, inhibition of DGAT1 enzymatic activity significantly reduced the production of

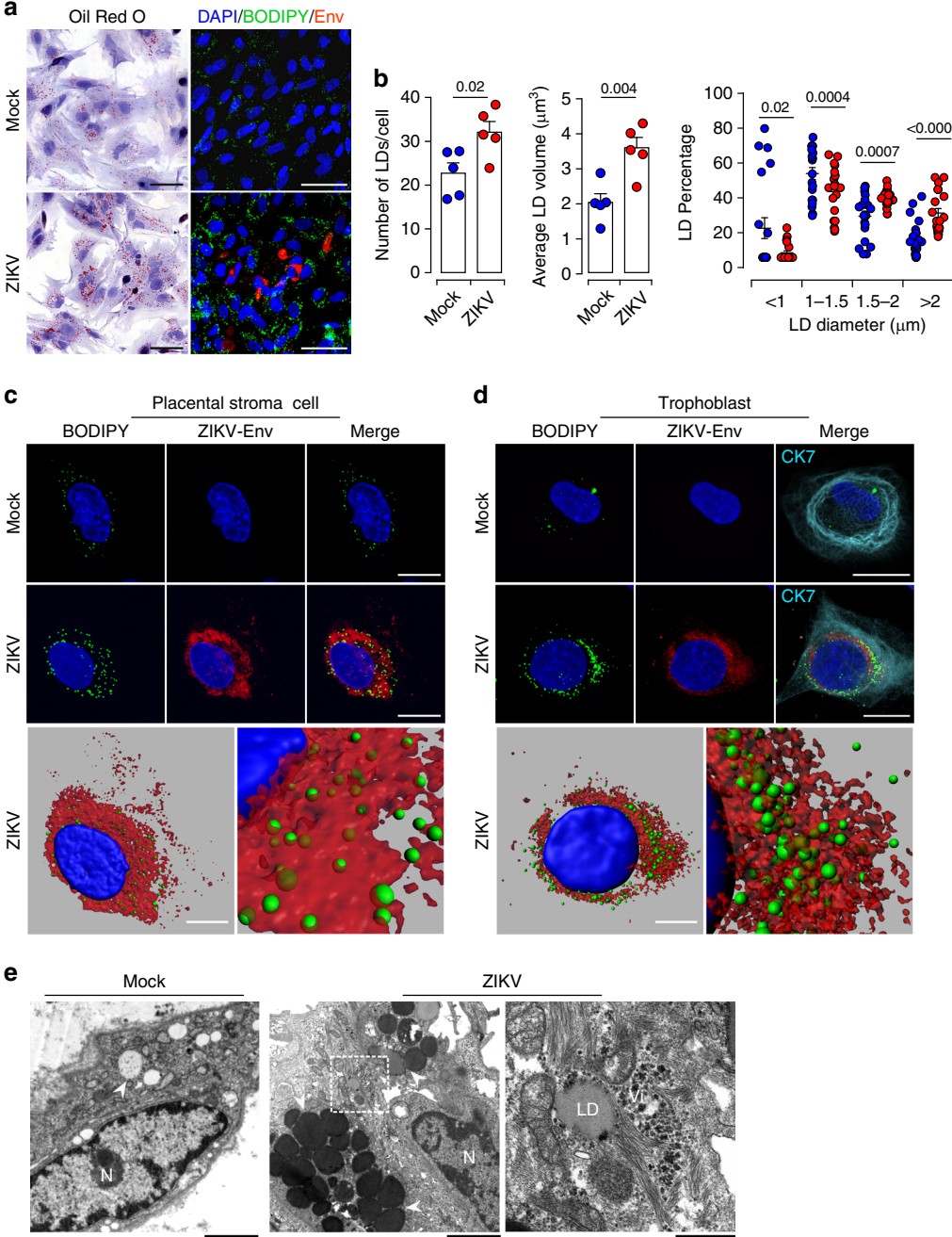

**Fig. 4 Alterations of LD morphometric features by ZIKV infection of primary placental cells.** Matched freshly isolated placental cells are mock- or infected with ZIKV at MOI of 1 for 5 days. **a** Oil Red O staining (Left panel). Co-immunostaining of LDs with the BODIPY 493/503 dye (green), anti-Flavivirus group antigen antibodies (Env, red), and nuclei (DAPI in blue) (Right panel). Scale bar 50 μm. **b** Quantification of LD number per cell, average volume and diameter distribution by Imaris software. LD morphometric features are analyzed in more than 400 cells for each placenta from five independent donors. P values are computed using two-tailed Student's t test. **c–d** Representative confocal images with high magnification of Z-stack projections. stromal cell (**c**), CK7 positive trophoblast cell (**d**). BODIPY 493/503 (green), Env (red), cytokeratin 7 (CK7 in cyan) and nuclei (DAPI in blue). Scale bar, 20 μm. Bottom panels showing 3D models and higher magnification focus on ZIKV-infected placenta stromal cell (**c**) and trophoblast cell (**d**). Scale bar, 10 μm. **e** Representative TEM micrographs showing LDs in (**a**) mock- and (**b**) ZIKV-infected placental cells (white arrowheads), nucleus (N). Scale bar, 2 μm. **c** High magnification of the white square delineated area in (**b**) with a concentration of LDs around a large amount of viral particles (Vi). Scale bar, 500 nm. Data are representative of at least three independent experiments. Source data are provided as a Source Data file.

infectious particles in a dose depend-manner. We observed more than 200xfold decrease in viral production with 5 μM DGAT1i treatment, compared to DMSO controls (Fig. 6d). Our findings demonstrate that specific blockade of DGAT1 enzymatic activity disrupts the biogenesis of LDs and thereby hinders the virus life cycle.

**ZIKV-infected cell exhibits intracellular membrane rearrangement.** Manipulating the distribution of lipids to build highly complex membranous webs, ROs, is a trademark of positive-stranded RNA viruses. The enrichment of lipid species with complex structures in infected placentas prompted us to investigate next whether ZIKV induces membrane remodeling to build

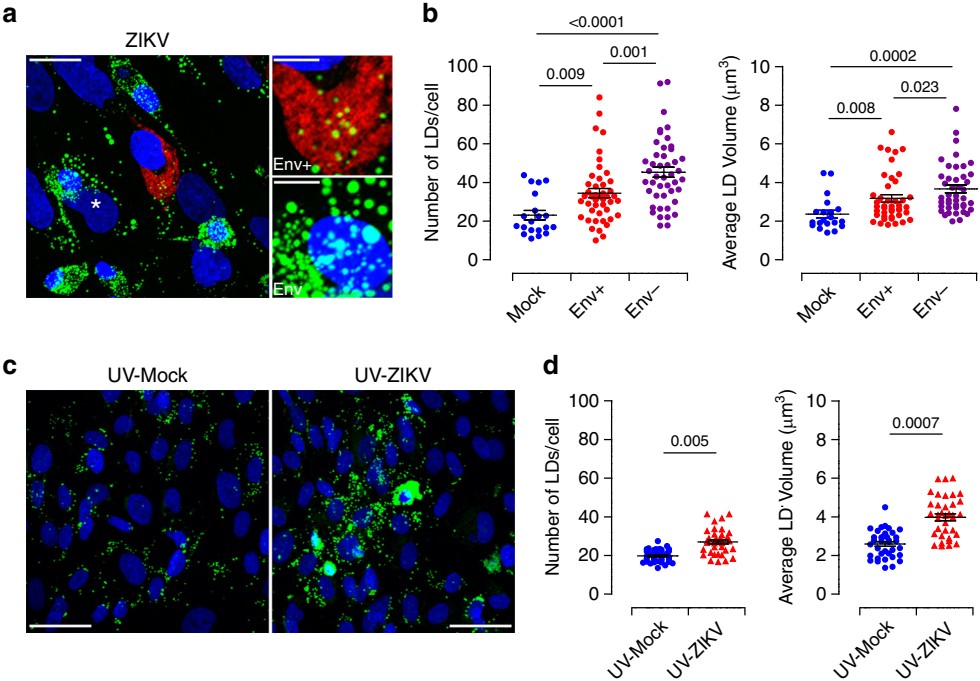

**Fig. 5 Bystander effect of ZIKV infection on LD biogenesis.** Placental cells were infected with ZIKV MOI of 1 for five days. **a** LD staining with the BODIPY 493/503 dye (green), anti-Flavivirus group antigen antibodies (Env) staining distinguishes ZIKV+ (red) from ZIKV− bystander cells (asterisk). High-magnitude images of LD are shown in the right panels. Scale bar, 20 µm. High magnification image insets showing LDs in both ZIKV+ (Env+) and ZIKV− (Env−) bystander. Scale bar, 5 µm. **b** Morphometric measurements of LD number and volume based on Z-stack projections of confocal images. Data from five different donors are shown as dot plot graphs. Each data point represents the average number of LDs per cell and average volume from mock (blue) and ZIKV-infected cells (red) or bystander neighboring cells (violet). Results are presented as mean values ± SEM and analyzed by one-way ANOVA, with Dunnett's multiple comparison test. **c** Supernatants from mock- or ZIKV-infected primary placental cells were collected from five independent donors 5 dpi and UV-irradiated. Freshly isolated placental cells are challenged with the UV-irradiated supernatants for 3 days. Immunofluorescence staining of LDs with the BODIPY 493/503 dye (green), ZIKV Env protein (red) and nuclei (DAPI in blue). Scale bar, 50 µm. **d** LD number and volume analysis by Imaris software. Each data point represents the average number of LDs per cells and average volume from Mock- (blue) and ZIKV+ cells (red). Results are presented as mean values ± SEM and analyzed by two-tailed Student's *t* test. Source data are provided as a Source Data file.

ROs. High-resolution TEM revealed the presence of convoluted membranes (CMs) and vesicle packets in the perinuclear region of infected trophoblast cells (Fig. 7a). A large number of viral particles were found within the ER lumen with an apposition of multiple viral-induced vesicular structures (Ve). These ZIKV-induced vesicular structures were heterogeneous in size (50–130 nm) but the majority of them were within a narrow diameter range of 90–110 nm (Fig. 7b). Similar structures housing viral RNA and proteins were previously found in Flavivirus infected hepatoblastoma cell line[39–41]. We also found spherical Ve either directly linked with viral particles or tightly connected with virion bags containing large clusters of assembled viral particles (Fig. 7c). Ve were connected with each other directly via pores (arrowhead) and occasionally with membrane tubes (T). These membrane structures are likely to convey cytoplasmic constituents to Ve and release of newly synthesized viral RNA in the cytoplasm as has been reported for West Nile virus (WNV)[39].

Mitochondrial and ER-associated membranes (MAM) are well-recognized sites for ATP synthesis, lipogenesis and export as well as for viral replication[14,42]. Viral particles were present within the ER lumen closely associated with mitochondria (Fig. 7d). We also found viral particles at the tip of the Golgi apparatus cisternae (arrow) indicating their transfer to the trans-Golgi network (TGN) for maturation. Elongated mitochondria were located near the Golgi apparatus (G). Finally, budding of progeny virions occurred through the genesis of microvesicles containing large clusters of viral particles (Fig. 7e), which are then released as

300–500 nm diameter bi-layer lipid membrane-surrounded extracellular vesicles (EV).

Together, these ultrastructure observations demonstrate that ZIKV provokes intracellular membrane rearrangements providing topological evidence of the placental differential metabolic reprogramming. To the best of our knowledge, this is the first report highlighting usurpation of the placental extracellular vesicular pathway by ZIKV for budding of progeny virions.

**ZIKV infection induces mitochondrial dysfunction.** Mitochondria are essential organelles involved in the sensing of metabolic homeostasis and communication with other cellular organelles such as LDs or peroxisomes. Alterations to placental mitochondria functions have been associated with compromised human pregnancies[43–45]. Therefore, we further addressed the impact of ZIKV infection on mitochondria functions. Mito-Tracker and ZIKV envelope co-staining was used to investigate the mitochondrial morphodynamics in mock- and ZIKV-infected placental cells. Compared to mock cells, where the mitochondria showed a classic distribution, the mitochondria network was completely disrupted by ZIKV infection and linear connected mitochondria intermingling with the Env-positive viral particles (Fig. 8a). 3D-modeling further highlighted the distortion of the mitochondrial network in ZIKV-infected cells. Further ultra-structure analysis using TEM showed long size mitochondria located near large clusters of viral particles (Fig. 8a). Compared to

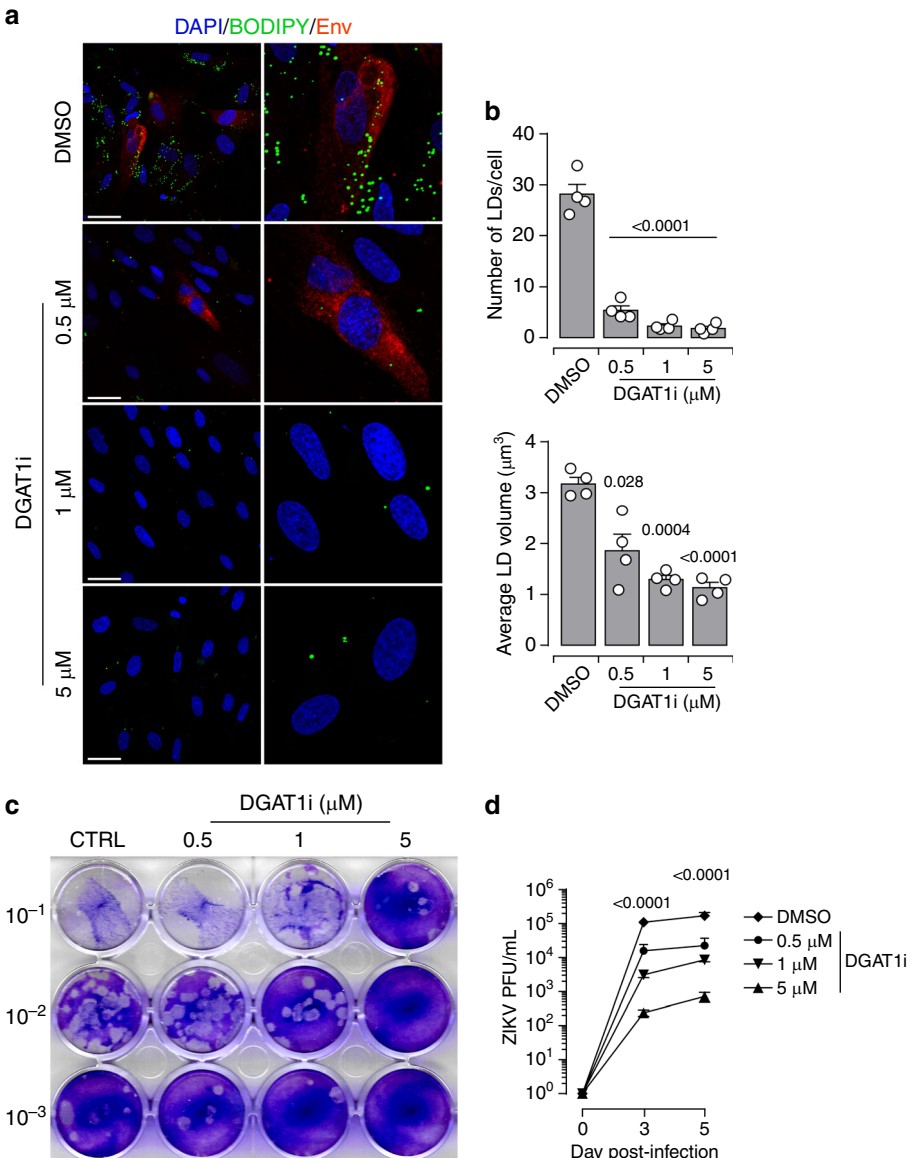

**Fig. 6 DGAT1 activity and LD formation are required for ZIKV life cycle.** Primary placental cells were pre-treated for 30 min with different concentrations of DGAT1i (0.5 μM, 1 μM, 5 μM), then cells were infected with ZIKV MOI of 1 up to 5 days in the presence of the DGAT1i. For the control, the cells treated with the same concentration of DMSO. **a** After ZIKV infection, the cells were co-immunostained for with the BODIPY 493/503 dye (green), envelope group antigen (Env in red), and nuclei (DAPI in blue). High-magnitude images are shown right panel, Scale bar 50 μm. **b** Quantification of LD amount per cell, the average volume of LD determined from 4 independent donors based on Z-stack projections of confocal images. Results are presented as mean values ± SEM. *P*-value denote comparisons between DMSO control and DTAG1i treated cells, two-tailed Student's *t* test. **c** Cell-free supernatants of cultures from growth curve experiments of DGAT1 inhibition assay were collected. The supernatants were 10-fold serial dilutions and 100 μl of diluted supernatant was incubated with Vero cells for plaque assay. **d** The kinetic of viral titers were determined based on plaque assay experiments. Data are presented as mean values ± SEM from three independent donors. Stars denote comparisons between DMSO control and DTAG1i treated cells. *P* values are computed using two-way ANOVA, with Tukey's multiple comparison test. Source data are provided as a Source Data file.

mock cells, the morphology of mitochondria was altered in ZIKV-infected placental cells (Fig. 8a, enlargement in the right panel).

To further investigate the mitochondrial function, we performed mitochondrial stress tests to probe the respiratory and glycolytic capacities using Seahorse Bioanalyzer. The analysis revealed that ZIKV infection inhibits oxygen consumption rate (OCR) but did not significantly change the extracellular acidification rate (ECAR) (Fig. 8b). Further quantitative analysis demonstrated that ZIKV infection considerably decreased the basal, maximal, and spare respiratory capacity (SRC) as well as the ATP-coupled OCR (Fig. 8c).

These observations reveal that ZIKV disrupts both the mitochondria morphology and respiratory capacity probably to impairment of the cell β-oxidation process, which in return will lead to metabolic alterations and accumulation of lipids within the placental tissue.

**Dual storm of inflammatory LMs in infected placentas.** Many viral pathogens interfere with the host lipid metabolism to promote the infection and associated inflammatory consequences[19,46,47]. LDs provide a distinct site for the production of specific LMs, Eicosanoids, which regulate inflammatory signaling cascades[18,48].

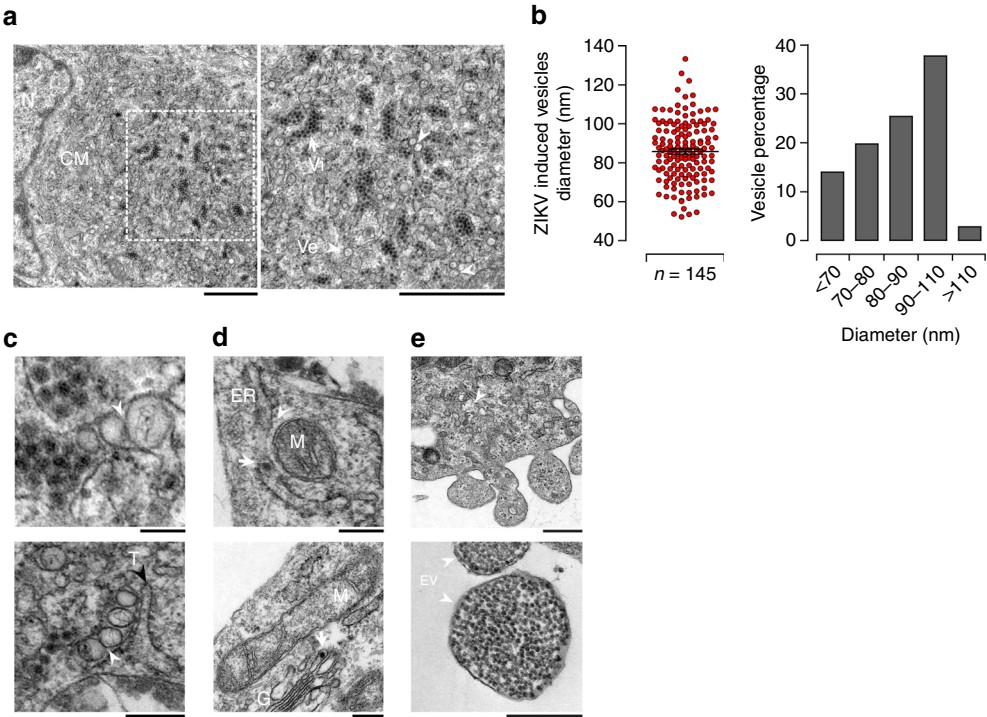

**Fig. 7 Ultrastructure analysis of ZIKV induces membrane rearrangements in placental cells. a** Representative TEM micrographs of 5 dpi ZIKV-infected placental cells. Convoluted membrane within ER lumen (CM) are localized next to the nucleus (N) shown in left panel. White rectangle denotes the enlarged region shown in right magnification panel. Large number of single membrane cisterna containing aggregated virions (Vi) which closely associated with virus-induced vesicles (Ve) are shown the right magnification panel. Arrowheads and arrows point to Ve and Vi, respectively. Scale bar, 1 μm. **b** Diameter of ZIKV-induced vesicles is shown in the left panel. The size distribution of Ve is shown in the right panel (n = 145) per ZIKV-infected placenta cells. Results are presented as mean values ± SEM. **c** Representative images of ZIKV-infected placental cells showing the localization of intracellular organelles depicted in ZIKV-infected cells (**a–f**). **a**, **b** Ve connected to each other through pores are indicated by white arrowheads. Scale bar, 100 nm. **c**, **d** Endoplasmic reticulum (ER) containing virions (Vi) are indicated by the white arrow, mitochondrion (M) and mitochondria-associated-membrane structures (white arrowhead) are shown, Golgi apparatus (G). **e**, **f** Representative micrographs showing viral budding from infected placenta cells and extracellular vesicles (EV) containing a large amount of viral particles are indicated with arrowheads. Scale bars, (**a–d**) 200 nm, (**e**) 500 nm, and (**f**) 200 nm. Data are representative of at least three independent experiments. Source data are provided as a Source Data file.

Within this context, we performed large-scale profiling of bioactive LMs to disclose the link between viral-induced lipid metabolism reprogramming and mitochondrial dysfunction. Among the profiled PUFA-derived LMs and their pathway metabolites, 25 mediators were detected in first-trimester placentas (Fig. 9). Although at different levels, many bioactive LMs were significantly up-regulated by ZIKV infection. In particular, significant increases were observed in those derived from the cyclooxygenase (COX), lipoxygenase (LOX) and cytochrome P450 enzymes (CyP450) pathways, as well as in HDoHE products of the omega-3 eicosapentaenoic acid (EPA) and docosahexaenoic acid (DHA) (Fig. 9a, b).

To look into the global inflammatory status of infected placentas, these 25 bioactive LMs were further clustered according to their pro-/anti-inflammatory properties and presented as a heatmap (Fig. 9c). ZIKV infection enhanced the production of both pro-/anti-inflammatory bioactive LMs. Despite the variability among donors, the level of pro-inflammatory prostaglandins PGE2 and PGE2α from the COX pathway and 5-Oxo-ETE from 5-LOX pathway were significantly increased in all tested placentas. Yet, this unbiased system approach also revealed that many LMs, known for their role in promoting the resolution of acute inflammatory responses, are also increased in the context of ZIKV infection. In particular, significant increases were observed for 12-HETE and 15-HETE from the 12-/15-LOX pathway, for the 8,9-EET of the CyP450 pathway as well as for the 14-HDoHE and 17-HDoHE bioactive products from DHA (Fig. 9a–c).

Accumulating evidence suggests that the occurrence of placental inflammation of either infectious or non-infectious origin can lead to placental dysfunction, and has been associated with obstetrical syndromes. Tilting the balance towards inflammatory response and regulating fetal trophoblast cell functions, might not only promote ZIKV survival but also impair the architecture of the developing fetal placenta. In support, Hematoxylin-Eosin (HE) staining revealed that while mock-infected placentas exhibited normal architecture of the chorionic villi, ZIKV-infected tissue showed significant disruption of the trophoblast layers evidenced by necrotic features such as pyknosis, karyorrhexis, and karyolysis (Supplementary Fig. 7a). Furthermore, we reviewed placenta from a congenital ZIKV case (10 weeks gestation). The placenta showed major changes in the architecture of the floating chorionic villi with a massive increase in CD68 positive Hofbauer cell number (Supplementary Fig. 7b).

Collectively, our findings evince a roadmap initiated by ZIKV reprogramming the lipid metabolome which would in turn induce disruption of the mitochondrial network and orchestrate a storm of immune mediators leading to fetal placental dysfunction.

## Discussion

The fetal placenta has an extraordinary energy need and is undoubtedly the most metabolically demanding organ. While changes to the placental metabolism have been clearly associated

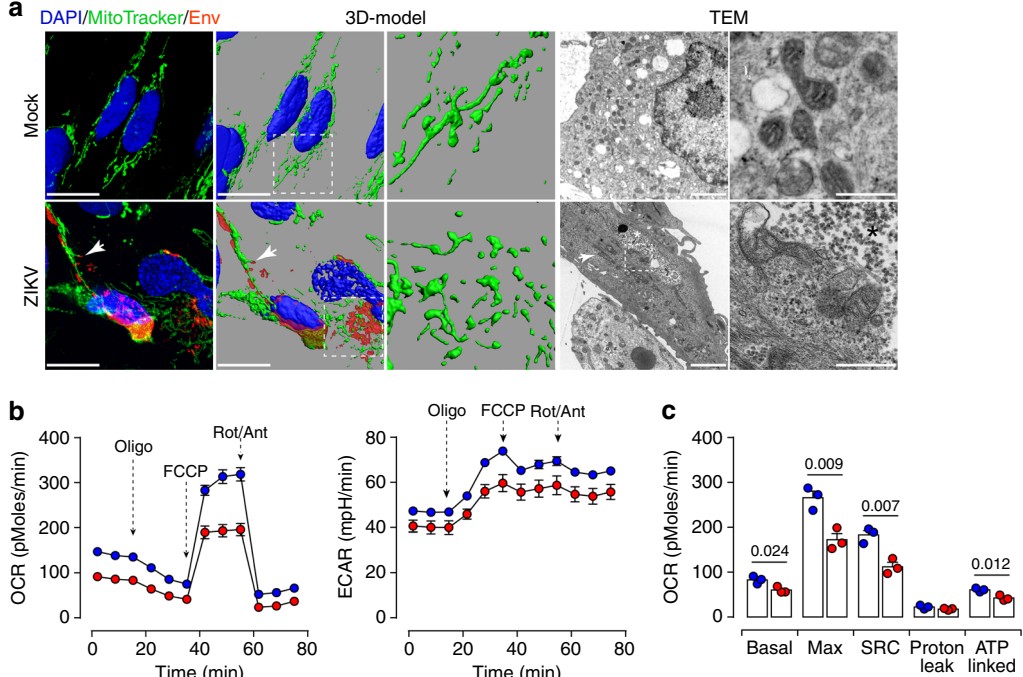

**Fig. 8 ZIKV infection impairs mitochondrial network distribution and respiratory capacities. a** Z-stack projection of confocal images from mock- and ZIKV-infected placental cells are stained with MitoTracker Deep Red FM (green) and anti-Flavivirus group antigen antibodies (Env, red). Nuclei are stained with DAPI (blue). 3D-view models are reconstructed by Imaris software. Scale bar, 20 μm. TEM micrographs showing the distribution in mock- and ZIKV-infected cells. Elongated size mitochondria are found in ZIKV-infected cells (white arrow). Scale bar, 2 μm. The white rectangle denotes the enlarged region shown in right magnification panels. Clustered viral particles (*) are found in close proximity with abnormal mitochondria. Scale bar, 500 nm. **b** OCR measured in mock- and ZIKV-infected placental cells by Seahorse XFp analyzer. **c** Bar graphs comparing OCR parameters: basal, maximal and spare respiratory capacities (SRC), and ATP-linked OCR. Data are presented as mean values ± SEM from three independent donors. *P* values are computed using two-tailed Student's *t* test. Source data are provided as a Source Data file.

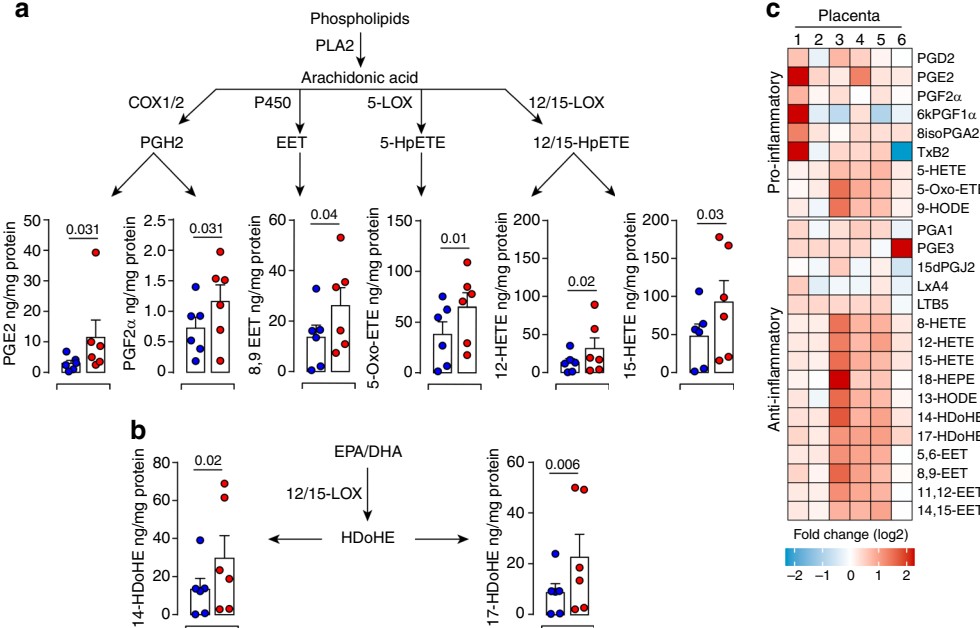

**Fig. 9 ZIKV-induced inflammatory response contributes to placental dysfunction.** Bioactive lipids are extracted from mock- and 5 dpi ZIKV-infected placentas and analyzed by LC-MS/MS. Absolute quantification of selected eicosanoids derived from PLs through the phospholipase A2 (PLA2) was calculated based on the calibration curves of deuterated internal standards and normalized by the amount of protein in each sample. Quantities of lipids are presented as ng/mg of protein for each placenta sample. **a** Quantification of eicosanoids derived from the cyclooxygenase (COX), lipoxygenase (LOX), and cytochrome P450 (CYP450) pathways. **b** Quantification of eicosanoids derived from the EPA-DHA. **c** Heatmap representing fold change of lipid mediators segregated according to their role as pro-/anti-inflammatory. Each column represents individual placentas from six independent donors (1–6). Results are presented as mean values ± SEM. *P* values are computed using paired two-tailed Student's *t* test. Source data are provided as a Source Data file.

with major pregnancy disorders, pre-eclampsia, FGR and GDM, their importance in the placenta's compensatory response to viral infections remains obscure. In this study, we made two primary contributions towards the understanding of ZIKV pathogenesis in early human pregnancy: (i) substantial adaptions of the fetal placental metabolism to meet the needs of viral replication and (ii) contextual associations between massive perturbation of the mitochondrial network, deep remodeling of the pro-/anti-inflammatory landscape and injury to the fetal placenta. To the best of our knowledge, this is the first study that links placental lipid metabolome and its associated inflammatory biomarkers to the degree of placental inflammation in ZIKV infection during pregnancy.

ZIKV infection triggered several adaptations to the placental lipid metabolism. These include enhancement of neutral lipids, specific PLs with long-fatty acid chains. During healthy pregnancy, lipids are commonly used for intracellular membrane biogenesis, energy storage directed down catabolic pathways or exported out of the placenta and become available to the developing fetus[23,49,50]. Here, we show that ZIKV-induced metabolic alterations impair the placental FA de novo synthesis and uptake pathway. Positive-stranded RNA viruses are able to mobilize FAs to increase lipid metabolism to build complex membrane structures for viral replication[51]. Our findings suggest that ZIKV-enhancement of FA transport/synthesis can be further stored in the LD core or processed to build ROs for the virus life cycle.

LDs are highly dynamic buffering organelles that regulate lipid homeostasis. Mature droplets engulfing sterol esters or TGs bud off the ER membranes to circumvent lipotoxicity in case of excessive amounts of lipids[52,53]. Many RNA viruses manipulate the LD biogenesis to favor viral genome replication and the production of progeny virions[54,55]. This is best exemplified by two members of the Flaviviridae family, HCV and DENV, which closely tangle with host cell LDs. Herein, ZIKV infection of placental tissue induces the accumulation of large size LDs. The growth of LD requires the integration of polar lipids, namely PLs and sterols, into the membrane layer and expansion of the hydrophobic central core by engulfment of neutral lipids. Our data clearly show that ZIKV infection increases DGAT1 expression further promoting the biogenesis and growth of the placental LDs. The essential role of these organelles in ZIKV infection is further support by our observation showing that interference with DGAT1 enzymatic activity suppresses the dynamic of LD growth and hampers viral replication. LDs entwining with viral replication factories could serve as platforms for the deposition and replication of the viral genome.

ZIKV-induced bystander accumulation of LD can be mediated through gap junction and soluble factors dependent mechanisms. It is very likely that modulation of the EV pathway and/or soluble factor release contribute to the observed bystander effect.

Thus, by programming the placental lipid metabolism, ZIKV coordinates the morphogenesis and functioning of key cellular organelles to establish a propitious environment for genome replication and virions assembly.

Recent reports suggest that similar to DENV, ZIKV might exploit the autophagy-dependent processing of LDs for efficient RNA replication[56,57]. In this sense, autophagy would then be a rapid and less selective mechanism for bulk processing of these lipids to supply building blocks for membrane biogenesis.

Flavivirus modify the lipid composition of the intracellular membrane structures. This is best supported by Flavivirus-induced enrichment of lipids that destabilize biological membrane curvature and permeability, and by the presence of a unique lipid repertoire in viral-induced ROs in infected cells[14]. Yet, in contrast to DENV and WNV[58,59], our comparative analyses of ZIKV-infected placentas did not reveal any alterations to

sphingolipid species, which underscores the tissue-specific particularity of ZIKV in regulating lipid metabolism.

Understanding how viruses regulate these topological changes is challenging. Our results show that ZIKV infection induces the formation of CMs within the ER lumen, which are in close proximity with an array of well-organized Ve. These vesicular structures, tightly interrelated with virus bags, are very likely to arise from dynamic mobility of ZIKV-induced CMs. While similar remodeling of ER membrane structures was previously observed in DENV and ZIKV infection[41,60], no functional link has yet been established between the ER membrane-derived vesicles and viral genome replication. Compared to published data on Huh-7 cell line, the mean diameter of ZIKV-induced Ve was much larger in primary placental cells (>90 nm), which suggests that cell-specific determinants could influence the morphogenesis of these ROs. In the future, it would be interesting to determine whether these size difference in ROs morphology reflects comparable with specific lipid composition.

Placenta-specific factors might contribute to shaping the landscape of intracellular membranes during ZIKV infection. In this regard, our structural observations would suggest that ZIKV RNA synthesis likely occurs within CMs, which are connected to viral-induced vesicles in placenta cells. These viral-induced vesicles may also participate in carrying newly assembled virions to TGN network for subsequent maturation as previously reported[61].

Mitochondria function is instrumental for many metabolic pathways. We provide pioneer evidence that ZIKV replication factories are in close proximity with MAM and that ZIKV infection disturbs the placenta mitochondrial network to meet its demanding replication needs. These ER-mitochondrial junctions constitute an excellent hub for coordinated lipid synthesis and trafficking as well as mitochondrial dynamics. Alterations to the tightly controlled mingling between these two organelles have been associated with many pathological conditions including pregnancy disorders[62]. Trafficking of ZIKV proteins to these highly dynamic membranes would not only help in the production of infectious progeny but also regulate placental intrinsic innate immune response. Mitochondria are also highly connected to LDs, allowing directional shuttling of fatty acids for subsequent mitochondria β-oxidation[63,64]. Proper partitioning and allocation of lipids between ER, LDs and mitochondria are therefore crucial to protect cells from lipotoxicity and cell death. By enhancing specific lipids, ZIKV may not only perturb the rate of lipid exchange and mitochondrial oxidative phosphorylation but also the crosstalk between these three essential hubs.

We demonstrate that the disruption of the mitochondrial network is linked to functional alterations illustrated by decreased oxygen consumption in ZIKV-infected placental cells. While ZIKV-induced mitochondrial stress may also contribute to a rise in placental lipid alterations, increased lipid content cannot be only explained by an increased synthesis/transport of FAs. The likely scenario rather includes alterations of the lipid esterification and downregulation in FA β-oxidation. We speculate that abnormalities in mitochondria engender excessive reactive oxygen species leading to oxidative stress, which can cause placental dysfunction. Since the placenta controls the quantity and composition of FAs delivery to the conceptus, even minor alterations of this mechanism would impair the delivery of lipids and lead to a higher risk of adverse birth outcomes. Furthermore, interrelated structure alterations MAM, mitochondria and LD dynamics observed in our ZIKV-infected placentas may also trigger massive inflammation hampering the success of pregnancy. Through large-scale analysis of bioactive mediators, we provide the first evidence that ZIKV perturbs the pro-/anti-inflammatory equilibrium of the placenta leading to tissue damage and massive

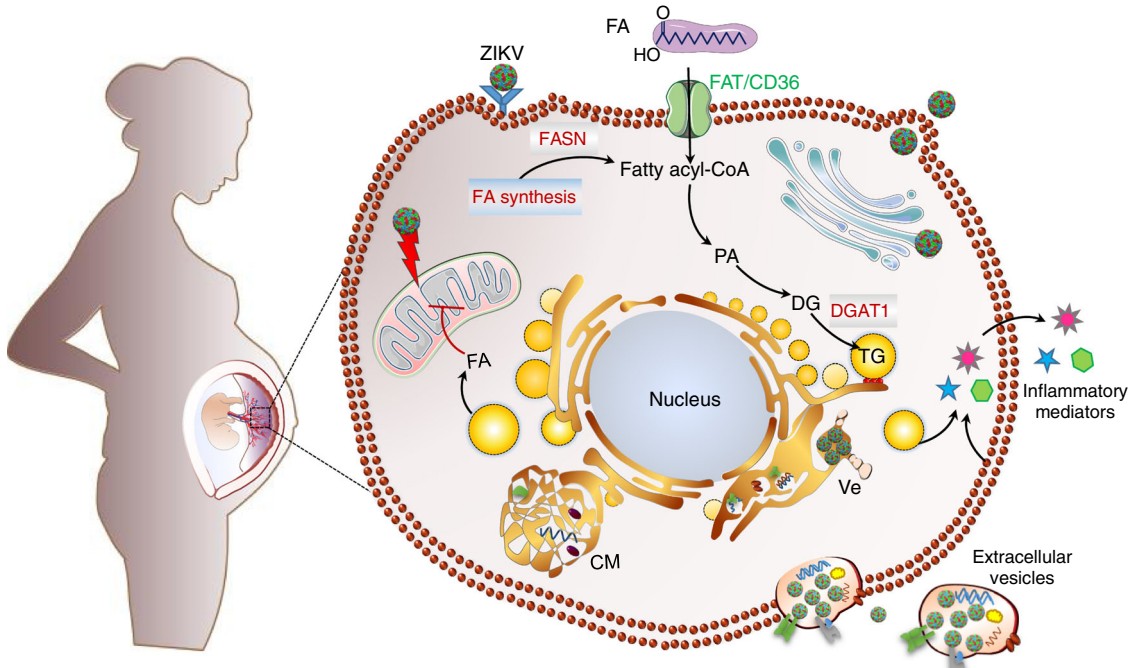

**Fig. 10 Summary of ZIKV-induced metabolic reprogramming of human placenta and specialized hubs that contribute to ZIKV life cycle.** By increasing placental lipid overload, ZIKV deprives the fetus of essential lipids and contributes to massive inflammation which may further impair the barrier function of the placenta.

infiltration of the villous core by inflammatory Hofbauer cells. While additional work is needed to fully define the link between mitochondrial network and accumulation of inflammatory metabolites at the molecular level, our study provides the first evidence linking lipid metabolome reprogramming to ZIKV infection drawing a potential path towards understanding viral-induced disruption of the placental barrier.

Collectively, reprogramming of the fetal placenta lipidome by ZIKV provides fundamental building blocks for optimal biogenesis and function of viral replication organelles. Damage to the mitochondria network may provide an amplification loop for placental inflammatory response. The metabolic disturbance can also interfere with placental intrinsic defense mechanisms allowing the virus to overcome the barrier function and reach the fetus (Fig. 10). Beyond illuminating novel mechanisms of ZIKV-induced pathogenesis during early pregnancy, our findings provide the groundwork for translational interventions directed towards supporting human pregnancy.

## Methods

**Patient samples.** Human sample collection for the study was approved by the French South-West & Outmer II ethical committee and registered at the French Ministry of Higher Education and Research (DC-2016-2772). Tissue samples were obtained with written informed consent from all participants in accordance with the Declaration of Helsinki guidelines.

**Tissue explant cultures and primary cell purification.** First-trimester placentas (7–11 weeks of gestation) were obtained from healthy women undergoing elective termination of pregnancy. About 0.3 cm$^2$ of tissue explant pieces were prepared from each placenta and kept in culture with DMEM:F12 (v:v) culture medium supplemented with 10% FBS. Primary cells were isolated from minced placental tissue subjected to enzyme digestion in culture medium containing 0.5 mg/mL collagenase IV (Sigma-Aldrich, France) and 0.1 mg/mL DNase I (Roche, France) for 25 min at 37 °C. Cell homogenates were then filtered (Thermo Fisher Scientific) and subjected to Ficoll-Hypaque density gradient (Amersham Biotech) separation, and then cultured in DMEM:F12 medium containing 10% FBS. For immuno-fluorescence and confocal microscopy analyses, cells were cultured on 12 mm glass coverslips (Knittel Glaser).

**ZIKV isolation and propagation.** ZIKV was isolated from sperm sample of immunocompetent Caucasian patient returning from Brasil[3,26–28]. The patient was tested HIV negative. Potential infection with dengue or chikungunya virus were ruled by ELISA Diapro (Diagnostic Bioprobes Srl, https://www.diapro.it) and RealStar Dengue and Chikungunya qRT-PCR (Altona Diagnostics, https://www.altona-diagnostics.com). Partial sequence of NS5 protein has been submitted to GenBank (KU886298). Sequence alignment and phylogenetic clustering attributed the isolated strain to the Asian lineage of ZIKV[26]. High titer stocks were obtained from early passaging in Vero cells(Vero ATCC® CCL-81™). Briefly, ZIKV was isolated from sperm sample was used to infect Vero cells. High titer supernatants were collected from 5 days post-infection (dpi). Viral titers were determined by RT-PCR and plaque assays on Vero cells. Comparative titration of viral stocks often shows a 2-log difference between RNA copy numbers and PFU. High titer viral stocks were stored as single use-aliquots and stored at −80 °C.

**Plaque assay.** Virus-containing medium from ZIKV-infected cells was collected at the indicated time post-infection. Confluent Vero cells grown in 12-well plates were infected with serial dilutions (up to 10$^{−6}$) of the virus-containing medium for 2 h, washed, and overlaid with 2% low-melting agarose with 1% FBS DMEM. The cells were stained with crystal violet 5 dpi. Plaques were counted and titers are calculated as PFU/mL.

**ZIKV infection of ex vivo tissue explants and primary cells.** Matched placental explants and purified primary placental cells were mock- or ZIKV-infected overnight at MOI of 1 in DMEM:F12 containing 2% FBS. After six washes in excess PBS, samples were cultured in complete medium containing 10% FBS. Supernatants were collected immediately after washing and every other day as indicated in the figures. Virus RNA was extracted from culture supernatant and viral RNA copies were detected using RealStar Zika virus RT-PCR kit 1.0 (Altona Diagnostics, https://www.altona-diagnostics.com).

**Quantification of ZIKV RNA.** After ZIKV infection, RNA was extracted from culture supernatants using MagNA Pure 96™ instrument (Roche, France). ZIKV RNA was detected by amplification of an NS5 fragment (RealStar Zika virus RT-PCR kit 1.0, Altona Diagnostic GmbH, Germany) using a Light Cycler 480 instrument (Roche Molecular Systems) according to the manufacturer's instructions. Internal standards were included to correct for potential variations in the amount of input material. Quantitative RT-PCR (qRT-PCR) values were further normalized and are given as viral RNA copy numbers/mL. Kinetic of viral production was determined as RNA copy number/mL, calculated based on a standard curve.

**Quantitative of gene expression by qRT-PCR.** For qRT-PCR analysis, total RNA from Mock- or ZIKV-infected primary placental cells were extracted using RNeasy

Kit (Qiagen) according to the manufacturer's instructions. Purified RNA was reverse transcribed using the SuperScript III first-strand synthesis system kit (Invitrogen). Gene expression was quantified by qRT-PCR using Light Cycler 480 SYBR green I master mix (Roche) and specific primers (Supplementary Table 1). qRT-PCR were performed in 96-well plates and run on a Light Cycler 480 instrument (Roche). The housekeeping hypoxanthine-guanine phosphoribosyl transferase (HPRT) was used as control. The results are presented as fold change from matched mock-infected controls calculated using the $2^{-\Delta\Delta CT}$ method.

**Lipid extraction and quantitative analysis by LC-MS/MS.** Five-day mock- or ZIKV-infected tissues were snap-frozen in liquid nitrogen and stored until extraction. Frozen tissue samples were crushed with a FastPrep®−24 Instrument (MP Biomedical, Santa Ana, CA) in 1 mL HBSS (Invitrogen, 14170112). After 2 crush cycles (6.5 m/s, 30 s), 50 μL were withdrawn for protein quantification. Different classes of lipids were extracted according to the manufacturer's instructions. Signal intensity for specific lipid species was extracted according to the retention time. Samples with spiked internal standards were analyzed by either gas/liquid chromatography followed by mass spectrometry as described in Supplementary methods. Representative chromatogram of different lipid species have been provided in Supplementary Figs. 8–10. The Mass Spectrometry conditions of different lipid species have been indicated in Supplementary Tables 2–5.

**Sample preparation for immunohistology.** For frozen section, placenta tissues were fixed in 4% paraformaldehyde overnight, rinsed in DPBS and infiltrated with 15–30% sucrose. Tissues were next mounted in optimum cutting temperature compound (OCT), snap-frozen and kept at −80 °C. Frozen tissues were cut into 5–10 μm sections using cryostat then processed for immunostaining. For classical IHC, congenital ZIKV-infected whole placentas were obtained from pathological termination of pregnancy (10 weeks of gestation). Tissues were fixed in 10% formalin and embedded in paraffin. Thin sections (3 μm-thickness) were processed for hematoxylin and eosin (HE) staining or immunostaining.

**Immunostaining.** For fluorescent staining, tissue sections were saturated with 10% normal serum and 5% bovine serum albumin (BSA) following incubated with primary antibodies, polyclonal antibody to ZIKV NS3 protein (1:200 dilution, GenTex, GTX133309), the trophoblast marker Cytokeratin 7 (CK7, 1:200 dilution, Dako M7018) overnight at 4 °C. The samples were then washed and incubated with Alexa Fluor 633 conjugated goat anti-mouse IgG1 (Invitrogen, A21126), Alexa Fluor 555 conjugated donkey anti-rabbit IgG (Invitrogen, A31572) secondary antibodies (1:500 dilution). LDs were stained with 10 μg/mL BODIPY 493/503 (4,4-difluoro-1,3,5,7,8-pentamethyl-4-bora3a,4a-diaza-s-indacene) (Molecular Probes, D3922). Confocal z-stacks were captured using LSM710 confocal microscope (Carl Zeiss, Germany) captured with 63x oil objective. Image processing was performed using ImageJ free software (version 1.5 J, NIH). Oil Red O staining was carried out in placenta tissue thin sections and placental cells. Staining was performed with 0.2% Oil Red O solution for 10 min, and nuclei were counterstained with hematoxylin solution (Sigma Aldrich). 6-μm-thick sections of paraffin-embedded from congenital samples were immunostained with the classical macrophage marker CD68 (1:100 dilution, Dako GA609). Images were taken with a Leica DMR microscope (Leica Microsystems, Nanterre, France) at 40x objective, using 3DHISTECH panoramic slide viewer (3DHISTECH Kft, Budapest, Hungary).

**Immunofluorescence and LD analysis.** Infected primary placenta cells grown on the coverslips were fixed for 20 min in 4% paraformaldehyde and permeabilized with 0.3% Triton X-100. Following, the cells were stained with anti-Flavivirus group antigen antibodies (D1-4G2-4-15, 1:400 dilution, Merck-Millipore, MAB10216), polyclonal antibody anti-ZIKV NS3 protein (1:300 dilution GenTex GTX133309), the trophoblast marker Cytokeratin 7 (CK7, 1:200 dilution, Dako M7018), incubated at 4 °C overnight. Alexa Fluor 555- goat anti-mouse IgG2a (Invitrogen, A21137) and Alexa Fluor 633 conjugated goat anti-mouse IgG1 (Invitrogen, A21126) secondary antibodies (1:500 dilution) were used for visualizing. After, cells were washed with PBS and stained with 4,6-diamidino-2-phenylindole (DAPI, 1:10,000 dilution, Sigma). For LD staining, fixed cells were incubated with 10 μg/mL BODIPY 493/503 dye (Molecular Probes, USA) for 30 min. Z-stacks images taken by LSM710 confocal microscope and processed using Imaris software (version 7.2.3, Bitplane AG). The total amount and, average volume and diameter of LDs were determined from ten randomly taken large field images from 5 independent donors.

**UV-irradiated conditioned medium.** Supernatants were collected from Mock- or ZIKV-infected primary placental cells 5 dpi, then UV-irradiated for 30 min using Spectroline EF-140/F UV lamp (220 volts, 50 HZ, 17 Amps). New fresh isolated placental cells were cultured for 3 days with UV-irradiated conditioned medium before immunostaining for LDs.

**DGAT1 inhibitors.** The pharmacological DGAT1 inhibitor (A922500) was obtained from MedChemExpress (MCE, USA). The inhibitor was well

characterized and published[65]. Cytotoxic effect of DGAT1 inhibitor (DGAT1i) on primary placental cells was determined by MTT assay (Sigma-Aldrich) and compared to DMSO control-treated cells. Cell viability was calculated as percentage of DMSO control-treated cells. For DGAT1 inhibition studies, primary placental cells were pre-treated for 30 min with different concentrations of DGAT1i, then challenged with ZIKV at MOI of 1. After 12 h exposure, cells were washed 6 times with fresh PBS before culturing in the presence of DGAT1i for 5 days. Supernatants were collected immediately after washing (0 dpi), 3 and 5 dpi. Viral titers were determined by plaque assay. Cells were processed for immunostaining of LD with the BODIPY 493/503 dye. The infection was visualized using the anti-Flavivirus group antigen antibody. LD number and size were determined using Imaris software.

**Electron microscopy.** The ZIKV-infected primary placental cells were fixed with 2.5% glutaraldehyde in 0.2 M cacodylate buffer for at least 1 h at room temperature. Then the cells were scraped from the flask and centrifuged at 3000g for 10 min. The pellets were fixed incubated in 1% osmium tetroxide for 1 h on ice. Fixed samples were dehydrated in a graded acetone series at room temperature and embedded in resin. Ultra-thin sections were prepared using microtome (Leica UC7, Germany) at 60 nm thickness and stained in 2% aqueous uranyl acetate and lead citrate. All images were then captured with a transmission electron microscope at 80 kV (Hitachi-7650, Japan).

**Mitochondrial morphology visualization and stress analysis.** Mitochondrial morphology was investigated using MitoTracker Deep red FM (Invitrogen, M22426) according to the manufacturer's instructions. Briefly, Mock- and ZIKV-infected placental cells were stained with 500 nM MitoTracker Deep red FM in condition medium for 30 min at 37 °C. Cells were then washed in fresh growth medium and fixed with 4% paraformaldehyde. After permeabilization with 0.3% Triton X-100, cells were incubated with anti-Flavivirus group antigen (D1-4G2-4-15, 1:400 dilution, Merck-Millipore MAB10216) at 4 °C overnight. Alexa Fluor 555- goat anti-mouse IgG2a (Invitrogen, A21137) secondary antibodies were used to visualize virus envelope protein. Images were captured by LSM710 Confocal microscopy.

Mitochondrial Stress test kit was applied to determine Oxygen Consumption Rate (OCR) and the Extracellular Acidification Rate (ECAR) using the Seahorse XFp analyzer (Seahorse Bioscience). Mock- and ZIKV-infected placental cells ($4 \times 10^4$ cells/per well) were seeded into plates at a cell density of $4 \times 10^4$ cells/well 12 h prior to analysis. Tissue culture medium was replaced by 180 μL of Seahorse XF Base medium. Cells were incubated in a $CO_2$-free incubator at 37 °C for 1 h to allow for temperature and pH equilibration before being loaded into the XFp analyzer. Under these basal conditions, 1 μM oligomycin, 2 μM FCCP fluorocarbonyl-cyanide phenylhydrazone or 1 μM rotenone and 1 μM antimycin A (Rot/Ant) were injected at the indicated time points. Mitochondrial OCR and ECAR were recorded for three cycles under basal conditions and following each sequential injection. Data were analyzed with Wave software (version 2.6.0, Seahorse Bioscience).

**Quantification and statistical analysis.** Statistical analyses were performed using GraphPad Prism software 7 (GraphPad Software). All data graphs were presented as the mean ± SEM. P values were determined by two-tailed Student's t test, one-way ANOVA, with Dunnett's multiple comparison test or two-way ANOVA, with Tukey's multiple comparison test. P-values of less than 0.05 are considered statistically significant and shown for each Figure. For heatmap analysis, data matrix normalized by autoscaling was exported into an Excel file and analyzed using R studio interface with an in-house R script. Red color indicates upregulation and blue indicate downregulation.

**Reporting summary.** Further information on research design is available in the Nature Research Reporting Summary linked to this article.

## Data availability
The authors declare that all the data supporting the findings of this study are available within the article and its Supplementary information files, or are available upon reasonable request to the lead author. The source data underlying Figs. 1, 2b–h, 3c, 4b, 5b, d, 6b, d, 7b, 8b–c, 9 and Supplementary Figs. 1, 2, 3, 4b–c, 5b, 6b are provided as a Source Data file.

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

## Acknowledgements

The authors are grateful to clinical staff from Paule de Viguier maternity hospital for the collection of placentas and, to the cell imaging, histopathology and lipidomic core facilities for technical assistance. We thank Prof. L.B. Zeng and Dr. W.Z. Liu for help with electron microscopy, Dr. X.H. Tong and N Cenac for lipid profiling, Prof. E. Bahraoui and Prof. J. Izopet for critical comments on the manuscript. This work was supported by recurrent funds from the Inserm-CNRS-University of Toulouse. Dr. Q. Chen is a recipient of the Chinese Scholarship Council and the French National Agency for AIDS Research PhD fellowships.

## Author contributions

Q.C. conducted the experiments, analyzed data, drafted and revised the manuscript, J.G. and H.E.C. contributed to data analysis. Y.J.F. contributed to statistical analysis using R-studio, A.E. and Q.G. contributed to sample preparation, G.C. provided clinical samples, R.D. drafted and approved the manuscript, N.J.F. designed and supervised the research, drafted and revised the paper. Funding acquisition was carried out by N.J.F and R.D.

## Competing Interests

The authors declare no competing interests.
