## [Peer Review File · Nature Communications]

Reviewers' comments:

Reviewer #1 (Remarks to the Author):

I am commenting on technical aspects of the lipidomics workflow only.

The authors need to include more details on their lipidomics workflow in several places in order for the quality of the analysis to be properly evaluated.

1. How was the tissue homogenised for all these measures? Where were standards bought?
2. There is no information on how the authors determine full lipid categories in their figures, e.g. what is CE, TAG mean specifically, is this a summary of peak area for several species and if so, which ones. Provide full details on this, including the mass values used for each species (this applies to all methods).
3. Nomenclature needs to be corrected to the LIPID MAPS naming conventions, e.g. use of / implies knowledge of position of FAs and should be replaced to `_`. Check carefully all naming is consistent.
4. Example chromatograms for lipids measured should be shown in supplementary.
5. Authors should include their criteria for defining a peak e.g. signal/noise and points across a peak required for a lipid to be considered real and not below LOD.
6. For phospholipids/ceramide/sphingomyelins, provide full detail of this extraction method, and the LC/MS method which is missing. Provide information on whether this was an MRM method and if so what transitions were used? MS settings for this method are missing.
7. For bioactive lipid quantitation, provide more detail on SPE method, e.g. volumes, washing and elution solvents, etc. Also, full details on which standards and how this was done need to be included, including quantitation method, and what are the MS conditions, and the transitions used, etc.
8. Figure legends need to be far clearer on how lipids were measured, esp when just called CE or TAG, etc, and it is not clear how this was done.
9. Was smoothing used, and if so, this should either be not used, or a clear justification for why and how it was used needs to be provided, showing that it does not change the data or outcomes measured (generally smoothing is not recommended).

Reviewer #2 (Remarks to the Author):

In this paper the authors have described Zika virus infection in placental samples from early human pregnancy. Using large scale metabolomics the authors show that there is extensive lipid metabolic reprogramming, accompanied by membrane rearrangements and mitochondrial dysfunction resulting in secretion of pro and anti-inflammatory mediators which contribute to tissue damage. The data presented in this study are interesting; especially characterisation of the placental cells as a model for Zika infection is certainly very useful. My primary concern is that this study lacks a hypothesis or a working model where the underlying mechanism is being addressed. It appears too descriptive at this point and seems to be a start for the purpose of addressing more in-depth questions in Zika infection. Given the descriptive nature of the data, it is not clear to me if there is a specific immune-metabolic pathway that contributes to Zika virus pathogenesis.

General comments:

First, the observations on altered lipid metabolism, membrane rearrangements and mitochondrial dysfunction are not specific to placental cells – this has been observed in other Zika virus susceptible cells, including monocyte-derived cells and hepatocytes. It is not clear from the way the manuscript is written, whether this is a specific phenomenon that results in placental damage and associated foetal complications. Also, if (as per the authors) the primary cause of pathogenesis is virus triggered lipid metabolic alteration, this should be supported by studies with atleast a set of inhibitors targeting some of the obvious lipid metabolic pathways, or genetic depletion/deletion of some of the key genes to demonstrate that this is so. These would be simple experiments where specific depletion of genes for DAG biosynthesis/Kennedy pathway would be tested in infected cells to measure pathogenesis with readouts as production of inflammatory mediators, LD generation etc.

Specific comments:

Figure 1b: Since the mock and infected samples are matched, it would be useful if the data points on the two columns were presented in that manner, so that individual increases were evident.

Figure 1d-i: Given the significant increase in DAG and downstream products (PC, PE) containing long chain FAs, is it possible to test/comment on whether viral replication compartments are enriched in these specific lipid species? Or are they being utilised for altered placental membrane architecture as the authors suggest?

Figure 2: Along the same lines as that of Figure 1, it would be useful to measure whether alterations in the specific enzymes in the Kennedy pathway/CDP-DAG pathway results in altered end products. These could be either enzyme abundance or activity in infected cells.

Figure 3: Since most flavivirus infections are accompanied by consumption of lipid droplets, it would be useful to perform LD staining as a function of time in the infected cells to quantitate whether consumption is followed by increased synthesis and storage of LDs or vice versa. What is the authors explanation for increased LDs in uninfected cells? Is it triggered by secreted cytokines or other inflammatory mediators?

Figure 5: Given the data on increased abundance and size of LDs in cells treated with conditioned UV-irradiated media, how can we exclude the possibility that the increases are an effect of immune response to infection and mediated through soluble cytokines or inflammatory regulators? It might be useful to treat these cells with a cytokine cocktail (or inflammatory mediators described in figure 8) to test their effect in LD abundance and volume.

Minor comments:

Line 90: The choice of first-trimester placenta needs to be better discussed. The sensitivity of placenta to ZIKA infection decreases along with the placental development (Arora, Sadovsky et al. 2017). It would be interesting to discuss the possible differences in lipid metabolic profiles of different placental cells upon ZIKA infection and its implication in ZIKA pathogenesis.

Page 9, line 210: should be vesicle "packets" instead of pockets.

Reviewer #3 (Remarks to the Author):

This intriguing manuscript by Chen et al. evaluates the impact of Zika virus (ZIKV) infection of the

human placenta on metabolic reprogramming including mitochondrial function and the lipid metabolome. The manuscript is primarily descriptive. Alterations in lipid content might influence disease pathogenesis, although this cannot definitively be known based on the results presented here.

1. Although there are numerous statistically significant differences in lipid abundance resulting from ZIKV infection, it is not clear whether these effects are meaningful in terms of viral replication or disease pathogenesis. Perhaps if there were a way to manipulate lipid levels it would be possible to demonstrate the significance of these phenomena during ZIKV infection. (In other words, even if it is not possible to determine how metabolic reprogramming occurs as a consequence of ZIKV Infection, it seems important to be able to prove whether this directly impacts disease pathogenesis. Perhaps an animal model exhibiting a genetic defects or alteration in a relevant metabolic pathway may be helpful here?)

2. Numerous flaviviruses including ZIKV are essentially pan-tropic and cause cell death. How much is virus- or DAMP-induced cell death contributing to the reported differences in lipid abundance in the placenta?

Minor:

1. Although mentioned in the methods, the results section also should include more details about the specific strain, virus dose, etc. The methods also should include passage history. This is critical information for interpreting results.

2. For qRT-PCR experiments, were samples normalized to a standard curve? Were copies per mL determined based on a genome equivalent?

3. Numerous text corrections are needed:

Line 38: *Flaviviridae* should be italicized

Line 44: Disagree with the statement that mechanisms of fetal pathogenesis are poorly understood. There is now a vast literature on the topic. Maybe instead this could be rephrased to say "incompletely understood."

Line 91: The text should be specific about the ZIKV strain used. Which Asian strain? What is the name and identifier? Was it deep sequenced to verify absence of other pathogens? How was it propagated?

Line 232: "pathogenesis" (typo)

Line 307: "placental" (typo)

Line 437: "health" placenta should be healthy control placenta

There are quite a few text errors (e.g. inappropriately capitalized words, spelling, and/or grammatical errors).

Point-by-Point Response to the Reviewers

Reviewer #1 (Remarks to the Author)

I am commenting on technical aspects of the lipidomics workflow only. The authors need to include more details on their lipidomics workflow in several places in order for the quality of the analysis to be properly evaluated.

1. How was the tissue homogenised for all these measures? Where were standards bought?

Author Response:

We appreciate the reviewer for pointing out that some technical details are not clear or missing. Each placenta sample was cut into several explants of 0.3 cm² (~200 mg). At the end of each experiment, tissue samples were snap-frozen then extracted as described in the revised manuscript (page 18, line 425) and supplementary experimental procedures (line 16). The final quantification of lipids is normalized to the protein levels as described in the supplementary experimental procedures (page 4). Information regarding internal standards has been added in supplementary procedures (page 2) “All the internal standards (IS) for neutral lipids were purchased from Sigma Aldrich. IS for phospholipids were purchased from Avanti Polar Lipids (Alabaster, LA, USA). Deuterated IS for bioactive lipids were purchased from Cayman Chemicals (Ann Arbor, MI, USA)”.

2. There is no information on how the authors determine full lipid categories in their figures, e.g. what is CE, TAG mean specifically, is this a summary of peak area for several species and if so, which ones. Provide full details on this, including the mass values used for each species (this applies to all methods).

Author Response:

We agree with the reviewer that information on quantification lipids should be provided. Quantification of different neutral lipid species was achieved by comparison of the peak area of individual species relative to their class-specific internal standards. Values are then normalized to sample protein levels. The relative abundance of TG represents the sum of the relative abundance of different subspecies in the TG group. This covers TG (16_16_16), TG (16_16_18), TG (16_16_18), TG (18_18_18), TG (18_18_20).

The relative abundance of CE represents the sum of relative abundance of different subspecies in CE group including CE (C16) and CE (C18).

Similar calculation modes were applied to determine the relative abundance of phospholipid species. All the details on how to determine lipid relative abundance have been added in the revised figure legends (Fig.1 and Fig.2).

The mass values used for each species have been provided in our supplementary tables (Table 2-5).

3. Nomenclature needs to be corrected to the LIPID MAPS naming conventions, e.g. use of / implies knowledge of position of FAs and should be replaced to . Check carefully all naming is consistent.

Author Response

The lipid nomenclature was corrected according to the reviewer’s suggestions using the classical nomenclature on the LIPID MAPS. We revised all figures and figure legends throughout the paper as suggested by the reviewer.

4. Example chromatograms for lipids measured should be shown in supplementary.

Author Response

In the revised version of the manuscript, we included example chromatograms of neutral lipid (Extended Supplementary Figure 1) and phospholipids (Extended Supplementary Figure 2). We also

included representative chromatographs of the bioactive mediators PGE2 and 17-HDoHE (Extended Supplementary Figure 3).

5. Authors should include their criteria for defining a peak. e.g. signal/noise and points across a peak required for a lipid to be considered real and not below LOD.

Author Response

Since neutral lipids and phospholipids were analyzed using a relative quantification approach, relative values above 10 mV cut-off were considered positive. All the details concerning our criteria for defining the peak definition have been published elsewhere^{1,2}.

For bioactive lipids, deuterated internal standards were chosen to fit different metabolite families. Absolute quantification was analyzed based on the calibration curves established from internal standards. A signal-to-noise ratio (s/n) over 10 was included in the analysis. Additional details on Machine settings and LOD for different compounds have been published elsewhere³.

This information is now included in the revised supplementary experimental procedures (page 4).

6. For phospholipids/ceramide/sphingomyelins, provide full detail of this extraction method, and the LC/MS method which is missing. Provide information on whether this was an MRM method and if so what transitions were used? MS settings for this method are missing.

Author Response

We have added all the details for the extraction of phospholipids/ceramide/sphingomyelins as well as the LC-MS/MS method in supplementary experimental procedures (pages 2-3).

For these lipid analyses, we used MRM method. Transition information is now provided in the revised version of the manuscript under supplementary Table 4.

7. For bioactive lipid quantitation, provide more detail on SPE method, e.g. volumes, washing and elution solvents, etc. Also, full details on which standards and how this was done need to be included, including quantitation method, and what are the MS conditions, and the transitions used, etc.

Author Response

For LC-MS/MS on bioactive mediators, we provided all the detail about how SPE method was carried out and about solvent reagents including dilution and volumes. These detailed methods for each extraction step are now presented in supplementary experimental procedures (page 3-4).

Deuterated internal standards were chosen to fit different metabolites families. Their name, source and amount are indicated in the revised supplementary experimental procedures (pages 3-4). Internal standard time segment, RT and SRM conditions are depicted in red in supplementary table 5.

Absolute quantification of each bioactive species was determined based on the calibration curve established using different dilutions of deuterated internal standards. Then data were normalized by per mg of protein from each sample. The quantification method has been indicated in the supplementary experimental procedures (page 4). In addition, all MS conditions with time segments, RT and SRM are shown in supplementary table 5.

8. Figure legends need to be far clearer on how lipids were measured, esp when just called CE or TAG, etc, and it is not clear how this was done.

Author Response

Figure legends for Figs. 1, 2 and 9 (former Fig. 8) have been edited and the necessary information about the measurement method of different lipid species has been indicated in the revised manuscript (lines 831-935, lines 849-852, 941-944).

9. Was smoothing used, and if so, this should either be not used, or a clear justification for why and how it was used needs to be provided, showing that it does not change the data or outcomes measured (generally smoothing is not recommended).

Author Response

No smoothing was used for our data analysis. We do agree with the reviewer that smoothing should be avoided.

Reviewer #2:

Summary of comments from the reviewer:

In this paper the authors have described Zika virus infection in placental samples from early human pregnancy. Using large scale metabolomics the authors show that there is extensive lipid metabolic reprogramming, accompanied by membrane rearrangements and mitochondrial dysfunction resulting in secretion of pro and anti-inflammatory mediators which contribute to tissue damage. The data presented in this study are interesting; especially characterisation of the placental cells as a model for Zika infection is certainly very useful. My primary concern is that this study lacks a hypothesis or a working model where the underlying mechanism is being addressed. It appears too descriptive at this point and seems to be a start for the purpose of addressing more in-depth questions in Zika infection. Given the descriptive nature of the data, it is not clear to me if there is a specific immune-metabolic pathway that contributes to Zika virus pathogenesis.

Author Response

In healthy pregnancy, the fetal placenta is able to develop in a finely balanced environment between inflammation and immune suppression. Localized inflammation within the maternal-fetal interface is necessary for fetal implantation and tissue remodeling. However, excessive production of inflammatory mediators can cause placental insufficiency. This has been illustrated in many of the great obstetrical syndromes such as preterm birth, growth restriction, gestational diabetes and pre-eclampsia which involve placental insufficiency. Furthermore, lessons learned from infectious agents associated with devastating developmental outcomes suggest that even subtle changes in maternal inflammation can lead to devastating outcomes.

Hence, the effect of ZIKV infection on the lipid metabolome and major changes of the pro- and anti-inflammatory equilibrium within the placenta may very well result in placental dysfunction.

We have now modified the text in the introduction to put forward our working hypothesis and working model (lines 74-76).

General comments:

First, the observations on altered lipid metabolism, membrane rearrangements and mitochondrial dysfunction are not specific to placental cells – this has been observed in other Zika virus susceptible cells, including monocyte-derived cells and hepatocytes. It is not clear from the way the manuscript is written, whether this is a specific phenomenon that results in placental damage and associated foetal complications. Also, if (as per the authors) the primary cause of pathogenesis is virus triggered lipid metabolic alteration, this should be supported by studies with at least a set of inhibitors targeting some of the obvious lipid metabolic pathways, or genetic depletion/deletion of some of the key genes to demonstrate that this is so. These would be simple experiments where specific depletion of genes for DAG biosynthesis/Kennedy pathway would be tested in infected cells to measure pathogenesis with readouts as production of inflammatory mediators, LD generation etc.

Author Response

We appreciate the reviewer's comment. During pregnancy, the placenta exhibits high metabolic activity. Lipid metabolites are not only participated in the placenta itself development but also can be transported

to the fetus circulation. Metabolic alterations have been clearly associated with placental dysfunction. All these metabolism alterations can affect the placental membrane fluidity and metabolites transport to the fetus and impair fetus development. To our knowledge, our study is the first report that ZIKV infection induces global metabolic to the placental. In addition, our study provides the topological evidence that the ZIKV infection induces lipid droplet biogenesis and viral replication complex formation which fosters viral replication. The mitochondria damage and inflammatory mediators' production can induce server tissue damage. Impairment of these cellular hubs function has been clearly associated with pregnancy disorders.

According to the reviewer's suggestion, we performed additional analysis to identify cellular pathways impaired during ZIKV infection, and which can lead to placental metabolic alterations. We analyzed the expression of key enzymes involved in fatty acid transport/synthesis and LD biogenesis. Our data revealed that ZIKV induces a significant increase in the expression levels of all three enzymes. Since we are dealing with primary placenta cells, we used chemical inhibition rather than genetic modulation. Our analysis demonstrates that pharmacological inhibition of DGAT1 enzymatic activity suppresses LD biogenesis and reduces ZIKV replication, further providing a proof-of-concept approach for new opportunity to block placental infection (Fig. 6).

Specific comments:

Figure 1b: Since the mock and infected samples are matched, it would be useful if the data points on the two columns were presented in that manner, so that individual increases were evident.

Author Response

We agree with the reviewer that it is clearer to present the data points per matched placenta sample. Therefore, graph in Fig. 1b was changed to depicted clearly matched placenta samples from mock- or ZIKV-infected conditions.

Figure 1d-i: Given the significant increase in DAG and downstream products (PC, PE) containing long chain FAs, is it possible to test/comment on whether viral replication compartments are enriched in these specific lipid species? Or are they being utilised for altered placental membrane architecture as the authors suggest?

Author Response

It is likely that increased in DG as well as the downstream phospholipids PC and PE with long acyl chain participate in membrane biogenesis supporting building of replication factories. ZIKV, similar to other (+) RNA viruses, has to create new membrane webs with unique lipid composition to bring together cellular and viral factors that are necessary for viral replication. Many (+) RNA viruses appear to be dependent on PE-enriched membranes for replication, and depletion of PE can significantly inhibit viral RNA replication. Due to the cone-shaped lipid of PE, their enrichment during viral infection will induce negative membrane curvature and promote the formation of viral replication complexes. Enhanced PC levels were also shown to accumulate at HCV and poliovirus replication sites, revealing a common feature among the group of (+) RNA viruses.

On the other hand, human placenta preferentially take-up long acyl chain fatty acids and regulate their delivery to the developing fetus for CNS development. Long acyl chain fatty acids are fundamental to membrane biogenesis during placental development and function. Furthermore, many physiological and biochemical adaptations that occur during pregnancy ensure availability of these fatty acids. Our observed alteration to specific phospholipids with long-chain fatty acid tails would impair the function of placenta and the channeling of fatty acids to the developing conceptus. Whist, the mechanisms of fatty acid transfer to the fetus are not clear, impaired placental delivery of fatty acids may constitute potential underpinning key to developmental outcomes with life-long health problems in congenitally afflicted infants. Therefore, by usurping essential fatty acids the virus can hamper also the placenta membranes.

In the future, it would be interesting to find out which lipids are specifically enriched in the replication sites and whether this would alter the membrane function of the placenta and/or deprive the fetus of essential lipids.

Figure 2: Along the same lines as that of Figure 1, it would be useful to measure whether alterations in the specific enzymes in the Kennedy pathway/CDP-DAG pathway results in altered end products. These could be either enzyme abundance or activity in infected cells.

Author Response

We thank the reviewer for the suggestion and reiterate our response to the general comments above. We performed additional analysis to identify cellular pathways impaired during ZIKV infection, and which can lead to placental metabolic alterations. We analyzed the expression of FASN, FAT/CD36 and DGAT1 key enzymes involved in the fatty acid synthesis, transport, and lipid droplet biogenesis. Our analysis revealed that ZIKV induces a significant increase in the expression these three enzymes. Our findings suggest increased synthesis/ transport of fatty acids can serve as building blocks for biogenesis of LD and virus replication organelles. These new data are shown in the revised Figure 2 (Fi. 2h).

Figure 3: Since most flavivirus infections are accompanied by consumption of lipid droplets, it would be useful to perform LD staining as a function of time in the infected cells to quantitate whether consumption is followed by increased synthesis and storage of LDs or vice versa. What is the authors explanation for increased LDs in uninfected cells? Is it triggered by secreted cytokines or other inflammatory mediators?

Author Response

Kinetics of viral replication revealed that ZIKV RNA can be detected 24h post-infection reaching a plateau at 5 dpi. Although the infection stayed high for several days, we decided to perform our lipid analysis on 5 dpi for optimal infection and to avoid artifacts from cytopathic viral effects or tissue decay. Nonetheless, we did perform analysis both at 3 days and five on isolated placental cells (Fig. 4 and Supplementary Fig. 3). Our IF analysis of LD revealed increased the LD size and number at both time points with even higher increases at 5 dpi. Furthermore, inhibition of the DGAT1-mediated LD formation further supports the essential role of these hubs in viral replication. Altogether, our data suggest that LDs can act as a scaffold for efficient production of infectious progeny virions.

Previous studies have shown that the dengue virus is able to deplete LD stores through an autophagy-dependent mechanism in Huh-7.5 cell line. Therefore, we consider the differences in virus-induced LD dynamics can be inherent the metabolic status of the type of cells. However, we cannot exclude that part of the LD stock is consumed to provide the necessary energy for the replication process.

The observed bystander effect was further characterized in freshly isolated placenta cells.

This effect can occur in cell-to-cell exchange of components or paracrine pathways via soluble factors. Our findings that UV-irradiated conditioned media can reproduce similar effects on non-infected cells favors a paracrine effect mediated through secreted key inflammatory mediators (lipid mediators, cytokine/chemokines resulting from placenta innate immune response and even extracellular vesicles needed for the crosstalk between other and fetus). However, we can exclude the effect of viral proteins and/or RNA in the absence of any replication. Future studies should identify what are the factors involved in ZIKV-induced bystander effect.

Figure 5: Given the data on increased abundance and size of LDs in cells treated with conditioned UV-irradiated media, how can we exclude the possibility that the increases are an effect of immune response to infection and mediated through soluble cytokines or inflammatory regulators? It might be useful to treat these cells with a cytokine cocktail (or inflammatory mediators described in figure 8) to test their effect in LD abundance and volume.

We agree with the reviewer that some of the pathways induced in our placental cells might be induced LD dynamic and expansion. Some of these inflammatory factors have been shown to induce lipogenesis. For instance, PGE2 from the COX pathway can induce the expression of a specific enzyme of lipogenesis and trigger lipid accumulation hepatocytes. At this point, we did not test the direct effect of specific lipid mediators or cytokine/chemokine on first trimester placenta samples

During early pregnancy, the level of prostaglandins is tightly regulated to establish a balanced e inflammatory microenvironment needed for the implantation of the embryo. Their upregulation can hamper implantation and placental development. Future targeted studies would establish whether a given bioactive mediator and/or cytokines/chemokine can induce LD dynamics in placental cells knowing that the balance of immune mediators is finely tuned in this tissue.

Minor comments:

Line 90: The choice of first-trimester placenta needs to be better discussed. The sensitivity of placenta to ZIKA infection decreases along with the placental development (Arora, Sadovsky et al. 2017). It would be interesting to discuss the possible differences in lipid metabolic profiles of different placental cells upon ZIKA infection and its implication in ZIKA pathogenesis.

Author Response

We agree with the reviewer and we amended the text and provided a justification for our choice to work with first trimester samples (lines 90-94). We also included the reference by Arora Sadovsky et al. 2017 (ref 37 of the revised manuscript).

Page 9, line 210: should be vesicle “packets” instead of pockets.

Author Response

The typo mistake has been corrected in the revised manuscript (line 246)

Reviewer #3 (Remarks to the Author):

This intriguing manuscript by Chen et al. evaluates the impact of Zika virus (ZIKV) infection of the human placenta on metabolic reprogramming including mitochondrial function and the lipid metabolome. The manuscript is primarily descriptive. Alterations in lipid content might influence disease pathogenesis, although this cannot definitively be known based on the results presented here.

1. Although there are numerous statistically significant differences in lipid abundance resulting from ZIKV infection, it is not clear whether these effects are meaningful in terms of viral replication or disease pathogenesis. Perhaps if there were a way to manipulate lipid levels it would be possible to demonstrate the significance of these phenomena during ZIKV infection. (In other words, even if it is not possible to determine how metabolic reprogramming occurs as a consequence of ZIKV Infection, it seems important to be able to be able to prove whether this directly impacts disease pathogenesis. Perhaps an animal model exhibiting a genetic defects or alteration in a relevant metabolic pathway may be helpful here?)

Author Response:

In our paper, we have shown that ZIKV infection reprograms placental cell metabolism. To illuminate cellular pathways responsible for ZIKV-induced alteration, we analyzed the expression of key enzymes involved in fatty acid synthesis, transport, and LD biogenesis. Our analysis revealed that ZIKV induces significant increase in the expression of FASN, FAT/CD36 and DGAT1 enzymes. These findings suggest that increased synthesis/transport of fatty acids can serve as building blocks for biogenesis of LD and virus replication organelles. More importantly, our new data demonstrating the efficacy of

chemical inhibitors of DGAT1 enzymatic activity in limiting ZIKV infection provide proof that LD can be used as a scaffold for viral replication.

Previous studies have provided evidence that alterations and overload of specific lipids in the fetal placenta are associated with placental insufficiency and severe pregnancy complications such as pre-eclampsia, preterm birth and fetal growth restriction.

Our data suggest that a safe manipulation of the placental lipids might be an alternative therapy to limit congenital infection and/or the occurrence of adverse pregnancy disorders.

Mouse models have proven useful to study several aspects of the ZIKV infection, including maternal-fetal transmission, tissue tropism, and mechanisms of virus replication as well as vaccine development. Human pregnancy differs from mouse pregnancy because of the unique fetal trophoblast invasion into the maternal *decidua basalis* as well as an increasing number of factors that are specific to human placentation. Nonhuman primates are phylogenetically closely related to humans. However, the use of nonhuman primates, particularly during gestation, raises major ethical issues. Despite the fact that many aspects of human placentation can only be addressed using human cells and tissues, it is very unlikely for us to access such a model.

2. Numerous flaviviruses including ZIKV are essentially pan-tropic and cause cell death. How much is virus- or DAMP-induced cell death contributing to the reported differences in lipid abundance in the placenta?

Author Response:

It is clear that oxidized lipids and DAMPs released by dead cells can change the inflammatory microenvironment further amplifying the viral effect. Our analysis of freshly isolated placental cells was performed 3dpi and 5dpi with a low multiplicity of virus. We did not observe any cytotoxic effect at these time points, yet we have a significant effect on LD morphology and growth dynamics. Although we cannot exclude that early release of mediators can act as DAMPs or PAMPs, we believe that the effect we are seeing is virus induced.

In addition, Ceramide is well known for its role as second messengers inducing cell apoptosis. But, we did not observe significant changes in the ceramide level under ZIKV infection.

Minor point

1. Although mentioned in the methods, the results section also should include more details about the specific strain, virus dose, etc. The methods also should include passage history. This is critical information for interpreting results.

Response

We agree with the reviewer that information on the virus could be given in the result section. Therefore, we did summarize to provide minimal information about the specific strain and the amount of virus in the result section. These changes are highlighted in (lines 92-94) of the revised result section "Placentas were challenged with the Asian strain (KU886298) of ZIKV at 6×10^{10} RNA copies/mL (equivalent to MOI of 1) obtained from second passage in Vero cells⁴⁻⁸. We also provided extensive information about the strain, propagation mode and included additional references in the revised methods (lines 436-443) (see detailed response below).

2. For qRT-PCR experiments, were samples normalized to a standard curve? Were copies per mL determined based on a genome equivalent?

Author Response

ZIKV RNA was detected by amplification of an NS5 fragment using RealStar Zika virus RT-PCR kit 1.0 (Altona Diagnostic GmbH, Germany). Quantification of viral production was determined as RNA copy number per mL, calculated based on a standard curve. All the details are given in the revised methods (lines 460-466)

3. Numerous text corrections are needed:

Line 38: Flaviviridae should be italicized

Author Response

Flaviviridae in Line 38 was changed to *Flaviviridae* (line 39)

Line 44: Disagree with the statement that mechanisms of fetal pathogenesis are poorly understood. There is a now a vast literature on the topic. Maybe instead this could be rephrased to say “incompletely understood.”

Author Response

We agree with the review that huge amount of work has been provided by several groups. Thus, the term ‘poorly understood’ has been replaced according to reviewer suggestion by the term ‘not fully understood’ (line 44)

Line 91: The text should be specific about the ZIKV strain used. Which Asian strain? What is the name and identifier? Was it deep sequenced to verify absence of other pathogens? How was it propagated?

Author Response

We reiterate our response to point 1. ZIKV was isolated from sperm sample of a 32-year-old immunocompetent Caucasian patient returning from Brasil (KU886298)⁴⁻⁸ The patient was tested HIV negative. Potential infection with dengue virus or chikungunya virus was ruled out by ELISA Diapro (Diagnostic Bioprobes Srl, <https://www.diapro.it>) and RealStar Dengue and Chikungunya qRT-PCR (Altona Diagnostics, <https://www.altona-diagnostics.com>). The partial sequence of NS5 protein has been submitted to GenBank (KU886298). Sequence alignment and phylogenetic clustering attributed the isolated strain to the Asian lineage of ZIKV.

Although the description of viral strain was given in the original submission under the method section, we, now provide additional information about the viral strain identification and virus propagation under the revised methods section.

High titer stocks were obtained from early passaging in Vero cells as previously described⁴. Viral titers were determined by plaque assays on Vero cells. High titer viral stocks were stored at -80°C in single-use aliquots.

Line 232: “pathlogenesis” (typo)

Line 307: “placental” (typo)

Line 437: “health” placenta should be healthy control placenta

There are quite a few text errors (e.g. inappropriately capitalized words, spelling, and/or grammatical errors).

Author Response

We do apologize for unintentional typo errors in the text. The whole paper was edited by a native speaker (English@work).

References

- 1 Chiappini, F. *et al.* Metabolism dysregulation induces a specific lipid signature of nonalcoholic steatohepatitis in patients. *Scientific reports* **7**, 46658, doi:10.1038/srep46658 (2017).
- 2 Stuani, L. *et al.* Stable Isotope Labeling Highlights Enhanced Fatty Acid and Lipid Metabolism in Human Acute Myeloid Leukemia. *International journal of molecular sciences* **19**, doi:10.3390/ijms19113325 (2018).
- 3 Le Faouder, P. *et al.* LC-MS/MS method for rapid and concomitant quantification of pro-inflammatory and pro-resolving polyunsaturated fatty acid metabolites. *Journal of chromatography. B, Analytical technologies in the biomedical and life sciences* **932**, 123-133, doi:10.1016/j.jchromb.2013.06.014 (2013).
- 4 El Costa, H. *et al.* ZIKA virus reveals broad tissue and cell tropism during the first trimester of pregnancy. *Scientific reports* **6**, 35296, doi:10.1038/srep35296 (2016).
- 5 Fourcade, C. *et al.* Viral load kinetics of Zika virus in plasma, urine and saliva in a couple returning from Martinique, French West Indies. *Journal of clinical virology : the official publication of the Pan American Society for Clinical Virology* **82**, 1-4, doi:10.1016/j.jcv.2016.06.011 (2016).
- 6 Mansuy, J. M. *et al.* Zika virus: high infectious viral load in semen, a new sexually transmitted pathogen? *The Lancet. Infectious diseases* **16**, 405, doi:10.1016/S1473-3099(16)00138-9 (2016).
- 7 Mansuy, J. M. *et al.* Peripheral Plasma and Semen Cytokine Response to Zika Virus in Humans. *Emerging infectious diseases* **25**, 823-825, doi:10.3201/eid2504.171886 (2019).
- 8 Mansuy, J. M. *et al.* Zika virus in semen and spermatozoa. *The Lancet. Infectious diseases* **16**, 1106-1107, doi:10.1016/S1473-3099(16)30336-X (2016).

REVIEWERS' COMMENTS:

Reviewer #1 (Remarks to the Author):

The reporting of lipids is sufficiently improved. The authors have included the requested information.

Can they amalgamate all the supplementary (the main supplementary, the new extended one, and the tables that include the assay information) into a single file as having three separate supplementary files was a bit confusing at times.

Reviewer #2 (Remarks to the Author):

The authors have adequately addressed the concerns raised - especially the additional data with DGAT1 inhibitor and expression profiles of lipid metabolic enzymes strengthens the involvement of LDs in viral pathogenesis. Although the authors discuss in the rebuttal the possibility of soluble mediators to induce changes in LD dynamics in a paracrine fashion, it would be important to include this qualifier in the discussion section of the manuscript.

Reviewer #3 (Remarks to the Author):

Although this remains a very intriguing manuscript, some major and minor questions remain. The authors' response and careful attention to the previous reviewer questions is greatly appreciated.

Major:

1. The manuscript is still primarily descriptive and would be much stronger if inhibitor findings could be verified under physiological conditions (e.g., in a mouse model of ZIKV infection in pregnancy). The points about differences in mammalian placentas are fine, but knowing whether the pathway matters under physiological conditions would still greatly strengthen of the argument. Unfortunately, it seems that the authors do not have access to such a model. Furthermore, administration of inhibitors of lipid metabolism in pregnancy would likely be a challenge, since these drugs would likely exhibit toxicity to a developing fetus. It would be helpful to assess their impact in vivo.

2. The response to the major question about DAMPs was reasonable and is well-taken.

Minor:

1. Line 94: Here, the manuscript states that placentas were infected with 6×10^{10} RNA copies/mL. This is an extremely high titer, and some additional explanation would be helpful in understanding why RNA copies were reported instead of plaque forming units. How many plaque forming units were there in this volume? Normally a Zika virus stock grown in Vero cells does not exceed 10^7 per mL or 10^8 plaque forming units per mL at the very most, unless large volumes of media were ultracentrifuged to generate a concentrated stock. Alternatively, perhaps this stock contains a very large number of RNA-containing particles that are not infectious. If so, this would also be important to report, as a large number of non-infectious particles may also have biological effects.

2. The data in Figure 6 are very helpful in establishing an effect of this pathway on ZIKV replication. However, in panels 6c and 6d and in the methods for these panels, it is stated that placental cells were challenged with ZIKV. Was the MOI 1 for all panels in the figure? What is the limit of detection? (If no plaques were detected at the eclipse phase, then the data at the eclipse

phase should be plotted at the limit of detection, which would typically be higher than 10^0 or 1 viral particle per mL.) If the MOI was 1, then I assume there was a large amount of infectious virus in the culture, which was then washed away prior to the eclipse phase. Is this correct? If so, it might be helpful to indicate this by showing on the Y-axis the amount of ZIKV per ml prior to washing at the time of inoculation, and then indicating the amount of ZIKV (limit of detection if no plaques) at the eclipse phase. Perhaps more details could be included here as well to aid in assessment of the findings?

Point-by-Point Response to the Reviewers

Reviewers' comments NCOMMS-19-27017A

Reviewer #1 (Remarks to the Author):

The reporting of lipids is sufficiently improved. The authors have included the requested information. Can they amalgamate all the supplementary (the main supplementary, the new extended one, and the tables that include the assay information) into a single file as having three separate supplementary files was a bit confusing at times.

Author Response:

We appreciate the reviewer comment and we agree with the fact that having supplementary data in three separate file might be confusing. Therefore, we have amalgamated all the supplementary data in a single PDF file.

Reviewer #2 (Remarks to the Author):

The authors have adequately addressed the concerns raised - especially the additional data with DGAT1 inhibitor and expression profiles of lipid metabolic enzymes strengthens the involvement of LDs in viral pathogenesis. Although the authors discuss in the rebuttal the possibility of soluble mediators to induce changes in LD dynamics in a paracrine fashion, it would be important to include this qualifier in the discussion section of the manuscript.

Author Response:

We thank the review for his/her comments. We amended the discussion section accordingly.

Reviewer #3 (Remarks to the Author):

Although this remains a very intriguing manuscript, some major and minor questions remain. The authors' response and careful attention to the previous reviewer questions is greatly appreciated.

Major:

1. The manuscript is still primarily descriptive and would be much stronger if inhibitor findings could be verified under physiological conditions (e.g., in a mouse model of ZIKV infection in pregnancy). The points about differences in mammalian placentas are fine, but knowing whether the pathway matters under physiological conditions would still greatly strengthen of the argument. Unfortunately, it seems that the authors do not have access to such a model. Furthermore, administration of inhibitors of lipid metabolism in pregnancy would likely be a challenge, since these drugs would likely exhibit toxicity to a developing fetus. It would be helpful to assess their impact in vivo.

Author Response:

We thank the reviewer for his/her comments and valuable suggestions. The additional experiments that we have performed significantly improve the quality of the manuscript. We agree with the reviewer, that our work would have been much stronger if the inhibitory data could be further validated *in vivo* using an animal model relevant to human pregnancy. Nonetheless, this would be beyond the scope of the actual manuscript and would require not only new experiments but also additional fundings. Even if DGAT1 inhibitors have been proposed to treat obesity-associated diseases, inhibiting lipid metabolism during pregnancy would very challenging.

2. The response to the major question about DAMPs was reasonable and is well-taken.

Minor:

1. Line 94: Here, the manuscript states that placentas were infected with 6×10^{10} RNA copies/mL. This is an extremely high titer, and some additional explanation would be helpful in understanding why RNA copies were reported instead of plaque forming units. How many plaque forming units were there in this volume? Normally a Zika virus stock grown in Vero cells does not exceed 10^7 per mL or 10^8 plaque forming units per mL at the very most, unless large volumes of media were ultracentrifuged to generate a concentrated stock. Alternatively, perhaps this stock contains a very large number of RNA-containing particles that are not infectious. If so, this would also be important to report, as a large number of non-infectious particles may also have biological effects.

Author Response:

Since we are using tissue explants, we cannot calculate the actual PFU. Comparative titration of our viral stocks shows a 2-log difference between RNA copy numbers and PFU. Since we cannot calculate the exact PFU when using tissue explants, we thus optimized our experimental conditions to the size of the explants using RNA copies rather than PFU (El Costa et al. 2016). However, we cannot exclude that our viral stocks may contain non-infectious viral particles. To avoid any confusion about this calculation method, we included this details in method section.

2. The data in Figure 6 are very helpful in establishing an effect of this pathway on ZIKV replication. However, in panels 6c and 6d and in the methods for these panels, it is stated that placental cells were challenged with ZIKV. Was the MOI 1 for all panels in the figure? What is the limit of detection? (If no plaques were detected at the eclipse phase, then the data at the eclipse phase should be plotted at the limit of detection, which would typically be higher than 10^0 or 1 viral particle per mL.) If the MOI was 1, then I assume there was a large amount of infectious virus in the culture, which was then washed away prior to the eclipse phase. Is this correct? If so, it might be helpful to indicate this by showing on the Y-axis the amount of ZIKV per ml prior to washing at the time of inoculation, and then indicating the amount of ZIKV (limit of detection if no plaques) at the eclipse phase. Perhaps more details could be included here as well to aid in assessment of the findings?

Author Response:

The MOI of 1 concerns **Fig. 6a,b**. For panels **c** and **d**, cells are exposed to the virus for 12 hours then washed 6 times in fresh PBS before culturing in complete medium for 5 days. Day 0 stands for supernatant collected immediately after washing. After extensive washing, we cannot detect any plaque forming units, we therefore plotted d0 as 10^0 on the Y axis. Even if our stock contains non-infectious particles, most of this would be washed away at day 0. Titration shown in Fig. 6d is designed to estimate the infectious particles produced by the cells after washing out free inoculum. Therefore, our curve starts at day 0.

We amended the methods section to include these details.